# Data Leakage in Federated Averaging

**Dimitar I. Dimitrov**                                     *to: dimitar.iliev.dimitrov@inf.ethz.ch*
*Department of Computer Science*
*ETH Zurich*

**Mislav Balunović**                                        *mislav.balunovic@inf.ethz.ch*
*Department of Computer Science*
*ETH Zurich*

**Nikola Konstantinov**                            *nikolahristov.konstantinov@inf.ethz.ch*
*ETH AI Center*
*Department of Computer Science*
*ETH Zurich*

**Martin Vechev**                                           *martin.vechev@inf.ethz.ch*
*Department of Computer Science*
*ETH Zurich*

**Reviewed on OpenReview:** *https://openreview.net/forum?id=e7A0B99zJf*

## Abstract

Recent attacks have shown that user data can be recovered from FedSGD updates, thus breaking privacy. However, these attacks are of limited practical relevance as federated learning typically uses the FedAvg algorithm. Compared to FedSGD, recovering data from FedAvg updates is much harder as: (i) the updates are computed at unobserved intermediate network weights, (ii) a large number of batches are used, and (iii) labels and network weights vary simultaneously across client steps. In this work, we propose a new optimization-based attack which successfully attacks FedAvg by addressing the above challenges. First, we solve the optimization problem using automatic differentiation that forces a simulation of the client's update that generates the unobserved parameters for the recovered labels and inputs to match the received client update. Second, we address the large number of batches by relating images from different epochs with a permutation invariant prior. Third, we recover the labels by estimating the parameters of existing FedSGD attacks at every FedAvg step. On the popular FEMNIST dataset, we demonstrate that on average we successfully recover >45% of the client's images from realistic FedAvg updates computed on 10 local epochs of 10 batches each with 5 images, compared to only <10% using the baseline. Our findings show many real-world federated learning implementations based on FedAvg are vulnerable.

## 1 Introduction

Federated learning (McMahan et al., 2017) is a general framework for training machine learning models in a fully distributed manner where clients communicate gradient updates to the server, as opposed to their private data. This allows the server to benefit from many sources of data, while preserving client privacy, and to potentially use these massive amounts of data to advance the state of the art in various domains including computer vision, text completion, medicine and others (Kairouz et al., 2021).

Unfortunately, recent works (Zhu et al., 2019; Zhao et al., 2020; Geng et al., 2021; Geiping et al., 2020; Yin et al., 2021) have shown how to compromise the privacy aspect of federated learning. In particular, they demonstrated that an honest-but-curious server can reconstruct clients' private data from the clients' gradient updates only - a phenomenon termed *gradient leakage*. While these attacks can recover data even for

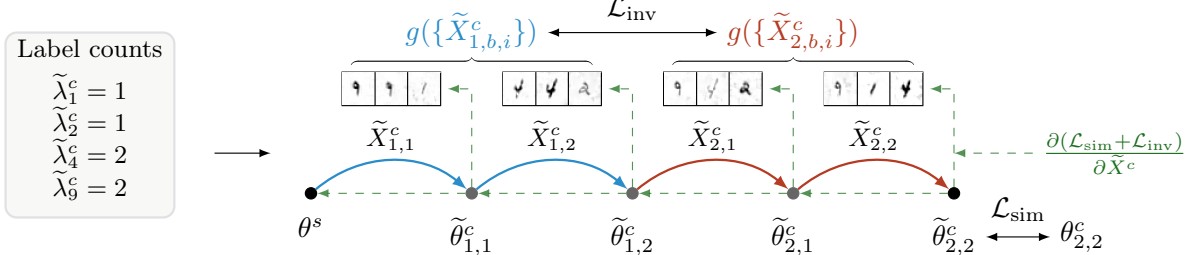

Figure 1: An overview of our attack on FedAvg. Our attack proceeds in two phases: first it estimates the label counts $\widetilde{\lambda}^c$ (left) and then uses these estimates to simulate the FedAvg updates on dummy inputs $\widetilde{X}^c$ (right). We then use automatic differentiation (green) to match the final weight $\widetilde{\theta}_{2,2}^c$ to the observed weight $\theta_{2,2}$ by minimizing the combination of the reconstruction loss $\mathcal{L}_{\text{sim}}$ and the epoch order-invariant prior $\mathcal{L}_{\text{inv}}$.

complex datasets and large batch sizes, they are still limited in their practical applicability. More specifically, they typically assume that training is performed using FedSGD where clients compute a gradient update on a single local batch of data, and then send it to the server. In contrast, in real-world applications of federated learning, clients often train the model locally for multiple iterations before sending the updated model back to the server, via an algorithm known as federated averaging (FedAvg) (McMahan et al., 2017), thus reducing communication and increasing convergence speed (Kairouz et al., 2021). Under FedAvg, the server only observes the aggregates of the client local updates and therefore, one may expect that this makes the reconstruction of client's private data much more difficult.

**This work: Data Leakage in Federated Averaging**  In this work, we propose a data leakage attack that allows for reconstructing client's data from the FedAvg updates, which constitutes a realistic threat to privacy-preserving federated learning. To this end, we first identify three key challenges that make attacking FedAvg more difficult than FedSGD: (i) during the computation of client updates, the model weights change, resulting in updates computed on weights hidden from the server, (ii) FedAvg updates the model locally using many batches of data, (iii) labels and parameters are simultaneously changing during the update steps.

These challenges are illustrated in Figure 1, where a client with a total of 6 samples trains the model for 2 epochs, using a batch size of 3 resulting in 2 steps per epoch. Here, the model weights $\theta$ change at every step and there are many different batches, with the samples being processed in a different order during the first (blue) and second (red) epoch.

Addressing these challenges, we introduce a new reconstruction attack that targets FedAvg. Our method first runs a FedAvg-specific procedure that approximately recovers the label counts $\widetilde{\lambda}^c$ within the client's data (left in Figure 1) by interpolating the parameters, thus addressing the last challenge. This information is then used to aid the second phase of the attack, which targets the individual inputs. To recover the inputs, our algorithm uses automatic differentiation to optimize a reconstruction loss consisting of two parts. The first part is a FedAvg reconstruction loss $\mathcal{L}_{\text{sim}}$, designed by simulating the client's training procedure, that accounts for the changing parameter values, addressing the first challenge. The second part solves the remaining challenge using an epoch order-invariant prior, $\mathcal{L}_{\text{inv}}$, which exploits the fact that the set of inputs is the same in every epoch, thus mitigating the issue of having a large number of batches during the local updates. To leverage this information, this term uses an aggregating order-invariant function $g$ to compress the sequence of samples in an epoch, and to then compare the results of this operation across different epochs.

We empirically validate the effectiveness of our method at recovering user data when the FedAvg algorithm is used on realistic settings with several local epochs and several batches per epoch on the FEMNIST (Caldas et al., 2018) and CIFAR100 (Krizhevsky et al., 2009) datasets. The results indicate that our method can recover a significant portion of input images, as well as accurate estimates of class frequencies, outperforming previously developed attacks. Further, we perform several ablation studies to understand the impact that individual components of our attack have on overall attack performance.

**Main Contributions**  Our main contributions are:

- Identifying key challenges in applying gradient leakage algorithms to FedAvg updates (Section 4.1).

- Solving these challenges through a novel algorithm based on automatic differentiation through a simulation of the FedAvg client update (Section 4.3) and an additional term incorporating prior information about the optimization process (Section 4.4), as well as new FedAvg-specific label reconstruction attack (Section 4.5).

- An implementation of this algorithm, that successfully attacks realistic FedAvg updates with multiple epochs and batches per epoch. Code is available at https://github.com/eth-sri/fedavg_leakage.

- A thorough evaluation of our attack on the FEMNIST and CIFAR100 datasets that validates its effectiveness at recovering private client data from FedAvg updates. In some of the settings we consider, our attack is able to accurately recover more than 50% of the clients' images, while prior methods are only able to recover 10%.

## 2   Related work

In this section we discuss closely related work in the area of data leakage for federated learning.

**Federated learning**  Federated learning (FL) was originally proposed by McMahan et al. (2017) as a framework for training machine learning models with private data from edge devices (or clients), without transferring the original data to a central server and instead communicating only gradient updates. Because of the potential of FL to enable training machine learning models at scale while respecting the sensitive clients' data, a substantial amount of recent research focuses on studying and enhancing the privacy guarantees provided by FL (Abadi et al., 2016; Bonawitz et al., 2017). However, recent work has cast doubt on the extent to which FL is inherently private, by demonstrating that in many cases the information about the model gradients alone is enough to reconstruct the clients' data.

**Gradient leakage**  Melis et al. (2019); Zhu et al. (2019) demonstrated that clients' data can be reconstructed from federated gradient updates. Next, Zhao et al. (2020) improved on the previous attack by showing that reconstructing inputs and labels separately simplifies the optimization process and leads to better reconstruction. Orthogonally, Zhu & Blaschko (2020) formulated gradient leakage attacks as a system of equations and provided an analytical solution under the assumption that the client has computed the update on a single data point. More recently, much work has been done on gradient leakage attacks on specific data domains through the use of input priors (Geiping et al., 2020; Dimitrov et al., 2022; Yin et al., 2021; Li et al., 2022; Deng et al., 2021). What these attacks demonstrate is that it is possible to reconstruct individual inputs from gradients aggregated over a batch of inputs, provided that a strong prior in the form of a regularizer is added to the optimization problem. Our proposed prior in Section 4.4 provides similar benefits. Additionally, following the observations of Zhao et al. (2020), Geng et al. (2021) designed attacks that approximate the frequency counts of labels in a single client's batch of data by looking at the gradients of the last linear layer of the model. We introduce a FedAvg-specific extension of this method in Section 4.5.

While the aforementioned works have achieved substantial progress in making gradient leakage attacks practical, the majority of these works focus on the reconstruction of a single data point or a single batch, given the respective gradient. Even works that consider FedAvg updates, such as Geiping et al. (2020) and Geng et al. (2021) focus on a setting where either small amount of epochs or small number of steps per epoch are executed on the clients' side. In contrast, we develop an end-to-end attack able to recover clients' data points from FedAvg updates computed on several epochs each with multiple intermediate steps – a problem of greater practical relevance given that FL clients usually perform many local iterations before sending their updates to the server to speed up convergence (McMahan et al., 2017).

**Defenses against Gradient leakage**  Several works have studied the task of defending against gradient leakage. There are three main types of defenses – heuristic (e.g Sun et al. (2021) and Scheliga et al. (2022)),

which are often broken by stronger attacks (Balunović et al., 2021); defenses based on adding random noise, that provide differential privacy guarantees (Abadi et al., 2016) but often result in reduced model accuracy; and defenses based on secure aggregation (Bonawitz et al., 2017) that ensure that only aggregated updates from multiple clients are disclosed to the server thus making the reconstruction harder. In Appendix C.1, we experiment with networks defended using Gaussian and Laplacian noise, as well as random gradient pruning first suggested by Zhu et al. (2019). Further, while attacking securely-aggregated updates is not the main focus of this work, following Wen et al. (2022), in Appendix C.5 we provide experiments on recovering data from aggregated FedAvg updates. As attacking aggregated updates from large number of clients remains unresolved problem in the gradient leakage community, future work is needed to obtain better results in this setting, however, we see even aggregated updates do not provide absolute client privacy.

**Other threat models**  In this work, we focus on the honest-but-curious server model for analyzing client's data privacy. That is, we assume that the server follows the standard training algorithm, but can attempt to reconstruct clients data through the received updates. Stronger threat models have also been studied. Works such as Fowl et al. (2022; 2021); Boenisch et al. (2021) demonstrate that the server can achieve much better reconstruction of private data instances by adapting the model weights and/or architecture.

## 3  Background

In this section, we review the classic components of federated learning, provide an overview of existing gradient leakage methods and their limitations, and introduce the relevant notation.

### 3.1  Federated learning

Throughout the paper, we assume that a server aims to train a neural network $f_\theta$ parameterized by a parameter vector $\theta$, with the help of $\mathcal{C}$ clients. We also assume that the neural network has $L$ layers of sizes $n_1, n_2, \ldots, n_L$ respectively, with the dimension of input variables being $d$ and the number of classes equal to $K$ (with $n_L = K$ for notational convenience).

The vast majority of FL methods used in practice are based on the FedAvg algorithm (McMahan et al., 2017). The optimization process consists of $T$ communication rounds. In each round, the server sends the current version of the model $\theta_t^s$ to a chosen subset of the clients. Each selected client $c$ then computes an update to the model, based on their own data, and sends an updated version $\theta_t^c$ of the model back to the server.

Algorithm 1 outlines the FedAvg's client update algorithm for a selected client $c$ at some communication round. We denote $X^c \in \mathbb{R}^{d \times N^c}$ and $Y^c \in [K]^{N^c}$ the client's private inputs and labels respectively, where $N^c$ signifies the number of private data points of that client and $[K]$ denotes the set $\{1, 2, \ldots, K\}$. At each local client's epoch $e \in \{1, 2, \ldots, \mathcal{E}\}$, the client randomly partitions its private data into sets of $\mathcal{B}^c = \lceil \frac{N^c}{m} \rceil$ batches $\{X_{e,b}^c \mid b \in [\mathcal{B}^c]\}$ and $\{Y_{e,b}^c \mid b \in [\mathcal{B}^c]\}$, each of size $m$ (Line 6). For each batch $b$ and epoch $e$, a local SGD update (Ruder, 2016) is computed for the cross entropy loss (denoted CE in Algorithm 1) on the respective batch of data. The SGD update is then applied to the network weights, producing the new intermediate weights $\theta_{e,b}^c$ (Line 8). Finally, the client sends the last updated weights $\theta^c = \theta_{\mathcal{E},\mathcal{B}^c}^c$ back to the server.

As a special case of FedAvg, when only one epoch ($\mathcal{E} = 1$) and one batch ($\mathcal{B}^c = 1$) are used locally in all rounds, one obtains the so-called FedSGD algorithm. While easier to attack, FedSGD is rarely used in practice, especially when the number of clients is large, as it leads to a significant increase in communication.

### 3.2  Gradient leakage attacks

In this section, we review existing attacks based on the honest-but-curious server model. The majority of these reconstruction methods, known as *gradient leakage attacks*, target the FedSGD training procedure (Zhu et al., 2019; Zhao et al., 2020), where the clients communicate a single gradient update to the server. As we attack FedAvg updates in this work, which share weights updates instead of individual gradients, we use the more general term *data leakage attacks* to refer to leakage attacks from both types of updates.

---

**Algorithm 1** The FedAvg's ClientUpdate Algorithm (Adapted from McMahan et al. (2017))

---

1: **function** CLIENTUPDATE($X^c$, $Y^c$, $f$, $\theta^s$, $\eta$, $m$, $\mathcal{E}$)
2:     $\mathcal{B}^c \leftarrow \lceil \frac{N^c}{m} \rceil$
3:     $\theta^c_{0,\mathcal{B}^c} \leftarrow \theta^s$
4:     **for** epoch $e \in [\mathcal{E}]$ **do**
5:         $\theta^c_{e,0} \leftarrow \theta^c_{e-1,\mathcal{B}^c}$
6:         $\{X^c_{e,b}\}, \{Y^c_{e,b}\} \leftarrow$ PARTITIONDATA($X^c$, $Y^c$, $m$)
7:         **for** batch $b \in [\mathcal{B}^c]$ **do**
8:             $\theta^c_{e,b} \leftarrow \theta^c_{e,b-1} - \eta \cdot \nabla_\theta \operatorname{CE}(f(X^c_{e,b}, \theta^c_{e,b-1}), Y^c_{e,b})$
9:         **end for**
10:    **end for**
11:    **return** $\theta^c_{\mathcal{E},\mathcal{B}^c}$
12: **end function**

---

**Reconstructions based on a single gradient**  To describe how existing attacks work, consider the case where the batch size $m = 1$, so that $\frac{\theta^s - \theta^c}{\eta} = \nabla_\theta CE(f(X^c_i, \theta), Y^c_i)$ is a gradient with respect to a single data point $(X^c_i, Y^c_i)$ of the client $c$. The most common approach for reconstructing $(X^c_i, Y^c_i)$ from the gradient is to search for an input-output pair, whose gradient most closely approximates the one that was sent back to the server. Given a distance measure $D$ in gradient space, this translates to the following optimization problem:

$$\underset{(x,y)}{\arg\min} \, D\left(\nabla_\theta CE(f(x, \theta^s), y), \nabla_\theta CE(f(X^c_i, \theta^s), Y^c_i)\right). \tag{1}$$

The distance measure $D$ is commonly chosen to be the $L_2$ (Zhu et al., 2019), $L_1$ (Deng et al., 2021) or cosine distance (Geiping et al., 2020). In the case of FedSGD, it is known that additionally using prior knowledge of the input distribution can improve the attacks (Geiping et al., 2020; Yin et al., 2021; Dimitrov et al., 2022), as it can guide the design of regularization terms that help in navigating the large optimization search space.

**Label reconstruction methods**  Previous work (Zhao et al., 2020; Huang et al., 2021) has shown that in the cases when the labels of the associated data points are known, the optimization problem in Equation 1 is easier to solve for $x$ by standard gradients-based optimization methods. For this reason, a common approach in the gradient leakage literature is to separate the process of recovering the clients' labels from that of the input reconstruction (Geiping et al., 2020; Geng et al., 2021; Yin et al., 2021).

Here we describe the procedure proposed by Geng et al. (2021) to reconstruct a client's label counts from FedSGD updates, which we later on adapt to the case of FedAvg. The paper assumes that the last layer of the network is linear, i.e. $f(X^c_i, \theta) = W_{\text{FC}} \cdot f_{L-1}(X^c_i, \theta) + b_{\text{FC}}$, where $W_{\text{FC}} \in \mathbb{R}^{n_{L-1} \times n_L}$ is a trained weight matrix, $b_{\text{FC}} \in \mathbb{R}^{n_L}$ is an optional bias term and $f_{L-1}(X^c_i, \theta)$ is the activation of the network for input $X^c_i$ at layer $L-1$. In addition to the linearity assumption, Geng et al. (2021) assume that the $K$ network outputs are converted into probabilities $p_k$ by applying the softmax function and are then fed to the cross entropy loss for training, as in Algorithm 1. We note that these assumptions hold for most neural network architectures.

Under these assumptions, Geng et al. (2021) shows that one can precisely compute the label counts $\lambda^c_k$ from FedSGD updates if the following quantities, computed on $\theta^s$, are known: (i) $\Delta_k W^s_{\text{FC}}$ – the sum of the client's gradients with respect to the CE loss of the weights in $W_{\text{FC}}$ that correspond to the $k^{\text{th}}$ layer output, (ii) $p^s_{i,k}$ – the softmax probability for class $k$ computed on $(X^c_i, Y^c_i)$ and (iii) $O^s_i$ – the sum of the neuron activations in layer $L-1$ computed on $(X^c_i, Y^c_i)$. While $\Delta_k W^s_{\text{FC}}$ can be computed by the server from the client update, $p^s_{i,k}$ and $O^s_i$ are unknown as they depend on the client's data. Instead, Geng et al. (2021) suggest to use approximations $\widetilde{p}^s_k$ and $\widetilde{O}^s$, computed by feeding the network at parameters $\theta^s$ with random dummy data, as opposed to the unknown client's data point. This results in the following approximation to the label counts:

$$\widetilde{\lambda}^c_k = N^c \cdot \widetilde{p}^s_k - \frac{N^c \cdot \Delta_k W^s_{\text{FC}}}{\widetilde{O}^s}. \tag{2}$$

A crucial limitation of these gradient leakage attacks is that they are tailored to FedSGD. As we show in the next section, both the input and the label reconstruction are significantly harder in the context of FedAvg, yielding previously developed attacks inapplicable.

---

**Algorithm 2** Overview of our attack

---

1: **function** ATTACK($f$, $\theta^s$, $\theta^c$, $\eta$, $m$, $\mathcal{E}$)
2:     $\nabla\bar{\theta}^c \leftarrow \frac{1}{\eta \cdot U^c}(\theta^s - \theta^c)$
3:     $\{\widetilde{\lambda}_k^c\} \leftarrow$ RECLABELS($\theta^s, \theta^c, U^c$)
4:     $\widetilde{Y}^c \leftarrow$ RANDORDER($\{\widetilde{\lambda}_k^c\}$)
5:     $\widetilde{X}^c \leftarrow$ RANDINIT($m$, $\mathcal{E}$)
6:     **while** not converged **do**
7:         $\widetilde{\theta}^c \leftarrow$ SIMUPDATE($\widetilde{X}^c$, $\widetilde{Y}^c$, $f$, $\theta^s$, $\eta$, $m$, $\mathcal{E}$)
8:         $\nabla\widetilde{\theta}^c \leftarrow \frac{1}{\eta \cdot U^c}(\theta^s - \widetilde{\theta}^c)$
9:         $\ell \leftarrow \mathcal{L}_{\text{sim}}(\nabla\bar{\theta}^c, \nabla\widetilde{\theta}^c) + \lambda_{\text{inv}} \cdot \mathcal{L}_{\text{inv}}(\widetilde{X}^c)$
10:        $\widetilde{X}^c \leftarrow \widetilde{X}^c - \eta_{\text{rec}} \cdot \frac{\partial\ell}{\partial\widetilde{X}^c}$
11:     **end while**
12:     **return** MATCHEPOCH($\widetilde{X}^c$), $\widetilde{Y}^c$
13: **end function**

---

# 4 Effective Reconstruction Attack on FedAvg

In this section, we first discuss the challenges FedAvg updates pose for data leakage attacks in Section 4.1. We then present an outline of our reconstruction attack in Section 4.2 and discuss how it addresses these challenges. Finally, for the rest of the section we discuss each of the elements of the attack in more details.

## 4.1 Challenges to Data Leakage in Federated Averaging

We first outline the key challenges for performing gradient leakage when training with FedAvg, making the leakage more difficult than in the case of FedSGD where these problems do not exist.

The most discernible challenge is that in FedAvg clients perform multiple local updates, each computed on intermediate model parameters $\theta_{e,b}^c$ that are unavailable to the server. If the number of local updates is large, the change in parameter values is likely to be large, making the data leakage problem substantially more difficult. We address this issue in Section 4.3.

The second important challenge is that FedAvg updates the model locally using many different batches of data. More concretely, the client updates the model for several local epochs, in each epoch performing several local steps, and in each step using a different batch of data. As the server only sees the final parameters, it does not know which batches were selected at each step. This is a more difficult case than FedSGD which performs its update using only a single batch of data. We address this issue in Section 4.4.

Finally, the third challenge is that labels and parameters are simultaneously changing during local updates. In practice, this makes it difficult to reconstruct label counts (necessary to reconstruct the inputs) via exiting FedSGD methods. In Section 4.5 we introduce our method for label reconstruction which is more robust w.r.t to changing batch sizes and number of epochs.

## 4.2 Federated Averaging Attack

We now present our attack, which aims to address the aforementioned challenges by reconstructing the clients' data from the original weights $\theta^s$, the client weights $\theta^c$ and client parameters $\eta$, $m$ and $\mathcal{E}$. We provide an overview of our method in Algorithm 2. First, we invoke a label reconstruction procedure to generate the reconstructed labels $\widetilde{Y}^c$ (Line 3–4) and then use $\widetilde{Y}^c$ as an input to an optimization problem that minimizes a reconstruction loss $\ell$ (Line 9) with respect to a randomly initialized (via RANDINIT) input set $\widetilde{X}^c$ (Line 5–11).

To address the challenge with label reconstruction, Algorithm 2 relies on our algorithm RECLABELS. This algorithm is an improved, more robust version of the method of Geng et al. (2021) discussed in Section 3.2 and tailored to FedAvg. The algorithm no longer directly estimates network statistics $\widetilde{p}_k^s$ and $\widetilde{O}^s$ from $\theta^s$ (as introduced in Section 3.2), but instead estimates the statistics for each $\theta_{e,b}^c$ by interpolating between the statistics at $\theta^s$ and the statistics at $\theta^c$.

For the input reconstruction, Algorithm 2 uses our novel FedAvg-specific reconstruction loss $\ell$ (Line 9), which consists of two components. The first component is the gradient similarity loss $\mathcal{L}_{\text{sim}}(\nabla\bar{\theta}^c, \nabla\widetilde{\theta}^c)$, which uses a server-side simulation of the client's updates via the SIMUPDATE method to obtain the approximated averaged update $\nabla\widetilde{\theta}^c$ (Line 8) and measures its similarity to the true averaged update $\nabla\bar{\theta}^c = \frac{1}{\eta \cdot U^c}(\theta^s - \theta^c)$, where $U^c = \mathcal{E} \cdot \mathcal{B}^c$ denotes the number of local update steps of the weights. This component aims to address the issue of having multiple hidden local steps by simulating the client's updates end-to-end. The second part is our epoch order-invariant prior $\mathcal{L}_{\text{inv}}(\widetilde{X}^c)$, which takes advantage of the fact that, even though the number of local steps $\mathcal{U}^c$ is large in realistic FedAvg settings, each input is part of exactly one batch per epoch. Finally, we reconcile our predictions by invoking MATCHEPOCH (discussed later in the section).

For simplicity, we present the optimization step in Algorithm 2 as an SGD update with a fixed learning rate $\eta_{\text{rec}}$. In practice, we instead rely on Adam (Kingma & Ba, 2014), as it was shown to perform better for solving gradient leakage problems (Zhao et al., 2020). The gradients $\frac{\partial \ell}{\partial \widetilde{X}^c}$, required by SGD and Adam, are calculated via an automatic differentiation tool – in our case JAX (Bradbury et al., 2018).

Over the next few sections we formally describe the components of Algorithm 2. In particular, we first discuss the reconstruction of inputs $\widetilde{X}^c$ from given label counts $\lambda_k^c$, as the inputs are typically of primary interest to the attacker. To this end, we describe our server simulation method SIMUPDATE in Section 4.3 and our epoch order-invariant prior $\mathcal{L}_{\text{inv}}(\widetilde{X}^c)$ in Section 4.4. Finally, in Section 4.5 we explain how to compute the estimated label counts $\widetilde{\lambda}_k^c$, thus creating a complete end-to-end attack.

### 4.3 Input Reconstruction through Simulation

As described in Section 4.1, a FedAvg client computes its gradient steps on the intermediate model parameters $\theta_{e,b}^c$ unknown to the server. These parameters can substantially differ from both $\theta^s$ and $\theta^c$ when the number of local steps $U^c$ is high. To address this issue, our attack relies on the method SIMUPDATE to simulate the client's updates from Algorithm 1 and thus generate approximations of the intermediate $\theta_{e,b}^c$'s.

An important consideration for our simulation is how to split the optimization variables $\widetilde{X}^c$, initialized at Line 5 and repeatedly updated at Line 10 in Algorithm 2, as well as the label counts $\lambda_k^c$ (or their reconstructions $\widetilde{\lambda}_k^c$ in the case of an end-to-end attack), into batches, as the client's original split is not available to the attacker. We fix the split randomly into $\mathcal{B}^c$ batches $\widetilde{Y}^c = \{\widetilde{Y}_b^c \mid b \in [\mathcal{B}^c]\}$ (via RANDORDER) before the optimization in Algorithm 2 begins and keep the same split for all $\mathcal{E}$ epochs throughout the reconstruction. This provides a stable input order throughout the reconstruction and making the optimization simpler. We do this for two reasons: (i) the original client split of the labels into batches is hard to reconstruct, and (ii) we experimentally found that most batch splits produce similar weights at the end of each epoch. For the optimization variables $\widetilde{X}^c$, we choose to split it into $U^c = \mathcal{E} \cdot \mathcal{B}^c$ separate variables $\widetilde{X}^c = \{\widetilde{X}_{e,b}^c \mid e \in [\mathcal{E}], b \in [\mathcal{B}^c]\}$, one for each batch $b$ and epoch $e$.

With these batch splits, SIMUPDATE executes a modified version of Algorithm 1. The two modification are: (i) The original partition of the data (Line 6) is replaced by our partitioning as described above, and (ii) we replace all occurrences of $X_{e,b}^c$ and $Y_{e,b}^c$ in Line 8 with $\widetilde{X}_{e,b}^c$ and $\widetilde{Y}_b^c$. The final reconstruction error is then calculated as the distance $\mathcal{L}_{\text{sim}}$ between the true average weight update $\nabla\bar{\theta}^c$ and the simulated update $\nabla\widetilde{\theta}^c$. In this paper, we rely on the cosine distance for $\mathcal{L}_{\text{sim}}$, i.e. $\mathcal{L}_{\text{sim}} = 1 - \frac{\langle \nabla\bar{\theta}^c, \nabla\widetilde{\theta}^c \rangle}{||\bar{\theta}^c||_2 ||\nabla\widetilde{\theta}^c||_2}$, as Geiping et al. (2020) showed it reconstructs data well in a wide range of challenging settings.

In addition to the approach described above, we also experimented with other variants. In particular, we investigated a version that uses $\mathcal{B}^c$ input optimization variables $\widetilde{X}^c = \{\widetilde{X}_b^c \mid b \in [\mathcal{B}^c]\}$ that are shared across all $\mathcal{E}$ epochs, as in Geiping et al. (2020). This change is applied at initialization time (Line 5 in Algorithm 2). Opposed to our algorithm (described above), the optimization problem now becomes more complicated, since the optimization variables enter into the computation of multiple local steps. Further, we considered a simulation where $\mathcal{B}^c = 1$, which is relatively precise when the number of batches that the client uses per epoch $\mathcal{B}^c$ is small, but significantly diverges for larger $\mathcal{B}^c$, as typically used by FedAvg. We compare to these variants in Section 5 and find that they perform worse, confirming the effectiveness of our method.

### 4.4 Epoch Order-Invariant Prior

The use of multiple local batches of data for several epochs in the FedAvg client update makes the optimization problem solved by Algorithm 2 harder compared to FedSGD. In particular, several reconstructions of each input have to be generated and combined, as the inputs are repeatedly fed to the network. Ideally, we want to enforce the property that all of the reconstruction variables corresponding to the same input at different epochs hold similar values. However, the correspondence between the variables across epochs is hard to establish as the client randomly splits its inputs into different sets of batches at each epoch. We make the observation that the value of any order-invariant function $g$ that takes all of the client data points as inputs has the same value at different epochs, assuming that each input is processed exactly once per epoch. To this end, during the simulation our prior encourages this property for every pair of epochs $(e_1, e_2)$, by penalizing a chosen distance function $D_{\mathrm{inv}}$ between the values of $g$ on the optimization variables $\widetilde{X}^c$ at epoch $e_1$ and $e_2$. Let $\widetilde{X}^c_{e,b,i}$ denote the $i^{\mathrm{th}}$ element in batch $b$ and epoch $e$ in $\widetilde{X}^c$. This results in the loss term:

$$\mathcal{L}_{\mathrm{inv}} = \frac{1}{\mathcal{E}^2} \sum_{e_1=1}^{\mathcal{E}} \sum_{e_2=1}^{\mathcal{E}} D_{\mathrm{inv}} \left( g(\{\widetilde{X}^c_{e_1,b,i} \mid b \in [\mathcal{B}^c], i \in [m]\}), g(\{\widetilde{X}^c_{e_2,b,i} \mid b \in [\mathcal{B}^c], i \in [m]\}) \right). \tag{3}$$

In Section 5.4, we experiment with several different order-invariant functions $g$ and show that the best choice is dataset-dependent. Further, we show that the $\ell_2$ distance is a good choice for $D_{\mathrm{inv}}$. In Line 9 of Algorithm 2, the prior is balanced with the rest of the reconstruction loss using the hyperparameter $\lambda_{\mathrm{inv}}$.

Once we have reconstructed the individual inputs at different epochs, we need to combine them into our final input reconstructions $\{\widehat{X}^c_i \mid i \in [N^c]\}$. We do this in MATCHEPOCH (Line 12 in Algorithm 2) by optimally matching the images at different epochs based on a similarity measure and averaging them. The full details of MATCHEPOCH are presented in Appendix A. In Appendix C.6, we experimentally show that our matching and averaging procedure improves the final reconstruction, compared to using the epoch-specific reconstructions, which aligns with the observations from previous work (Yin et al., 2021).

### 4.5 Label Count Reconstruction

As we point out in Section 2, state-of-the-art FedSGD attack methods aim to recover the client's label counts $\lambda^c_k$ before moving on to reconstruct the inputs $X^c$. This is commonly done by computing statistics on the weights and the gradients of $f$ (Yin et al., 2021; Geng et al., 2021). As demonstrated by our experiments in Section 5.2, existing FedSGD methods struggle to consistently and robustly reconstruct the label counts $\lambda^c_k$. We postulate that this is due to the simultaneous changes across local steps of both the labels inside the processed client's batch and the network parameters. To address the challenge of recovering label counts robustly (to different batch and epoch sizes), we make the observation that the model statistics used by these methods often change smoothly throughout the client training process.

As the label counts reconstruction mechanism demonstrated in Geng et al. (2021) is the current FedSGD state-of-the-art, we chose to focus on adapting it to the FedAvg setting using the observation above. Recall from Section 3.2 that the method depends on the estimates $\widetilde{p}^s_k$ and $\widetilde{O}^s$ of the average softmax probabilities $p^s_k$ and of average sum of neuron activation at layer $L-1$. For FedSGD, both estimates are computed at $\theta^s$ and with respect to a dummy client dataset, since the true data of the client is unknown to the server. However, in the case of FedAvg, multiple local steps are taken by the client, with each batch using different labels and changing the intermediate model parameters (both labels and parameters remain unobserved by the server). To account for these changes, we compute approximate counts $\widetilde{\lambda}^c_{k,i}$ for each local step $i \in [U^c]$ and aggregate these counts to obtain the approximate label counts $\widetilde{\lambda}^c_k$ for the full dataset of the client.

We proceed as follows. First, we compute approximations $\widetilde{p}^s_k$ and $\widetilde{O}^s$ at $\theta^s$ (using a client dummy dataset) and approximations $\widetilde{p}^c_k$ and $\widetilde{O}^c$ at $\theta^c$ (using the same client dummy dataset), as described in Section 3.2. Then, we linearly interpolate these estimates between $\widetilde{p}^s_k$ and $\widetilde{O}^s$ and $\widetilde{p}^c_k$ and $\widetilde{O}^c$ for the $U^c$ steps of local training, that is, we set $\widetilde{p}^c_{k,i} = \frac{i}{U^c}\widetilde{p}^s_k + \frac{U^c-i}{U^c}\widetilde{p}^c_k$ and $\widetilde{O}^c_i = \frac{i}{U^c}\widetilde{O}^s + \frac{U^c-i}{U^c}\widetilde{O}^c$ for each $i \in [U^c]$ and $k \in [K]$. Next, we use these approximations to obtain approximate label counts $\widetilde{\lambda}^c_{k,i}$ for every training step $i \in [U^c]$, using

Equation 2. To compute the final approximate counts, we set $\widetilde{\lambda}_k^c = \frac{1}{\mathcal{E}} \sum_{i=1}^{U^c} \widetilde{\lambda}_{k,i}^c$. Note that we may need to adjust $\widetilde{\lambda}_k^c$ in order to enforce the invariant that $\sum_{k=1}^{K} \widetilde{\lambda}_k^c = N^c$. We remark that this interpolation of the local statistics is general and can be applied to other label reconstruction methods.

## 5 Experimental Evaluation

In this section, we present an experimental evaluation of our proposed attack and various baselines.

**Experimental setup**   We conduct our experiments on two image classification datasets. One is FEMNIST, part of the commonly used federated learning framework LEAF (Caldas et al., 2018). FEMNIST consists of $28 \times 28$ grayscale images of handwritten digits and letters partitioned into 62 classes. We evaluate with 100 random clients from the training set and select $N^c = 50$ data points from each. The other dataset is CIFAR100 (Krizhevsky et al., 2009), which consists of $32 \times 32$ images partitioned into 100 classes. Here, we simply sample 100 batches of size $N^c = 50$ from the training dataset to form the individual clients' data.

We ran all of our experiments on a single NVIDIA RTX 2080 Ti GPU. The runtimes of different methods and datasets are shown in Appendix D alongside with a discussion on the computational complexity of SIMUPDATE. We apply our reconstruction methods on undefended CNN networks at initialization time. We provide additional experiments on defended and trained networks in Appendix C. In all experiments, we assume the attacker has knowledge of $\mathcal{E}$, $m$, and $N^c$. We relax this assumption in the experiments in Appendix C.4. Throughout the section, FEMNIST attacks that use our epoch prior set the order-invariant function $g$ to the mean of the images in an epoch, while CIFAR100 attacks set it to the pixelwise maximum of the randomly convolved images. We investigate different choices for $g$ in Section 5.4. We provide the exact network architectures and hyperparameters used in our experiments in Appendix B.

The rest of the section is organized as follows. First, in Section 5.1, we focus on image reconstruction with known label counts. In this setting, we compare our attack to using non-simulation-based reconstruction methods and justify our algorithm design choices by showing that our method compares favorably to a selection of baselines. Next, in Section 5.2, we experiment with label counts recovery and show that compared to Geng et al. (2021) our method is more robust while achieving similar or better label reconstruction results. In Section 5.3, we evaluate our end-to-end attack that reconstructs both the inputs and label counts. Finally, in Section 5.4 we justify our choice for $g$ for the different datasets.

### 5.1 Input Reconstruction Experiments

In this section, we compare the image reconstruction quality, assuming the label counts per epoch are known (but not to which batches they belong), with methods from prior work (Geiping et al., 2020; Geng et al., 2021) as well as variants of our approach: (i) **Ours (prior)**, our full input-reconstruction method, including the order-invariant prior $\mathcal{L}_{\text{inv}}$, as described in Section 4.4, as well as the simulation-based reconstruction error $\mathcal{L}_{\text{sim}}$ that uses separate optimization variables for the inputs at different epochs, (ii) **Ours (no prior)**, same as **Ours (prior)**, but not using the epoch prior $\mathcal{L}_{\text{inv}}$, (iii) **Shared** (Geiping et al., 2020), the approach proposed by Geiping et al. (2020) which assumes same order of batches in different epochs in SIMUPDATE, as described in Section 4.3 (allows sharing of optimization variables without using the prior), (iv) **FedSGD-Epoch**, this variant simulates the FedAvg update with a single batch per epoch ($\mathcal{B}^c = 1$) and thus no explicit regularization is needed, (v) **FedSGD** (Geng et al., 2021), the approach proposed by Geng et al. (2021), disregards the simulation, and instead reconstructs the inputs from the average update $\nabla \bar{\theta}^c$ like in FedSGD.

To deal with the unknown label batch split at the client, for all attack methods we pick the same random batch split in Line 4 in Algorithm 2. We report the results for different values of the number of epochs $\mathcal{E}$ and batch sizes $m$ in Table 1, resulting in different number of local client steps $U^c$. For all experiments, we consider a reconstruction on FEMNIST (CIFAR100) successful if the corresponding PSNR value is $> 20$ ($> 19$) and we report two measures: (i) the percentage of images that were successfully reconstructed by the attack, and (ii) the average peak signal-to-noise ratio (PSNR) (Hore & Ziou, 2010), which is a standard measure of image reconstruction quality, between the client and the reconstructed images. We note that PSNR is on a logarithmic scale, so even small differences in PSNR correspond to large losses in image quality.

Table 1: The effectiveness of different methods for reconstructing images with known labels. We measure the percentage of successfully reconstructed images and the average PSNR of the reconstructions.

| Dataset | $\mathcal{E}$ | $m$ | $N^c$ | $U^c$ | Ours (prior) | | Ours (no prior) | | Shared | | FedSGD-Epoch | | FedSGD | |
|---|---|---|---|---|---|---|---|---|---|---|---|---|---|---|
| | | | | | Rec (%) | PSNR | Rec (%) | PSNR | Rec (%) | PSNR | Rec (%) | PSNR | Rec (%) | PSNR |
| FEMNIST | 1 | 5 | 50 | 10 | **42.3** | **19.7** | 42.3 | 19.7 | 42.3 | 19.7 | 37.7 | 19.5 | 37.7 | 19.5 |
| | 5 | 5 | 50 | 50 | **63.1** | **21.2** | 56.2 | 20.7 | 59.0 | 20.6 | 56.1 | 20.4 | 13.7 | 17.4 |
| | 10 | 10 | 50 | 50 | **63.8** | **21.3** | 58.5 | 20.9 | 62.0 | 20.7 | 60.2 | 20.6 | 11.8 | 17.2 |
| | 10 | 5 | 50 | 100 | **65.5** | **21.5** | 50.6 | 20.3 | 59.9 | 20.3 | 56.0 | 19.6 | 8.3 | 16.7 |
| | 10 | 1 | 50 | 500 | **49.5** | **20.1** | 21.5 | 16.7 | 14.9 | 16.1 | 1.6 | 4.5 | 5.2 | 16.6 |
| CIFAR100 | 1 | 5 | 50 | 10 | **9.1** | **16.2** | 9.1 | 16.2 | 9.1 | 16.2 | 7.7 | 16.1 | 7.7 | 16.1 |
| | 5 | 5 | 50 | 50 | **55.2** | **19.2** | 53.0 | 19.1 | 10.8 | 16.4 | 10.9 | 16.3 | 1.2 | 14.2 |
| | 10 | 10 | 50 | 50 | **62.7** | **20.3** | 60.1 | 20.0 | 23.8 | 17.2 | 26.2 | 17.4 | 0.6 | 13.8 |
| | 10 | 5 | 50 | 100 | **58.8** | **19.8** | 56.4 | 19.6 | 15.7 | 16.4 | 8.5 | 15.0 | 0.4 | 13.2 |
| | 10 | 1 | 50 | 500 | **12.2** | **15.4** | 11.0 | 15.2 | 0.4 | 12.5 | 0.0 | 8.9 | 1.3 | 12.8 |

For the methods which produce multiple reconstructions of the same image (the two **Ours** methods and **FedSGD-Epoch**), we use matching and averaging approach, described in Section 4.4, to generate a single final reconstruction. We use a linear assignment problem similar to the one in Appendix A to match these final reconstructions to the client's original images before computing the reported PSNR values. All results are averaged across all images and users. In Appendix C.3, we further experiment with the setting where both the label counts and their batch assignments are known.

**Evaluating reconstructions** From Table 1 we make the following observations. First, the full version of our method provides the best reconstructions in essentially all cases. Second, we find that our epoch prior $\mathcal{L}_{\text{inv}}$ improves the results in almost all cases except for the case of a single client epoch ($\mathcal{E} = 1$). We point out that in this case the three methods **Ours (prior)**, **Ours (no prior)** and **Shared**, as well as the two methods **FedSGD-Epoch** and **FedSGD**, are equivalent in implementation. We observe that the use of the prior results in a bigger reconstruction gap on the FEMNIST dataset compared to CIFAR100. Our hypothesis is that the reason is the added complexity of the CIFAR100 dataset. In Section 5.4, we further demonstrate that more complex order-invariant functions $g$ can help close this gap. Additionally, we observe that using separate optimization variables in SimUpdate sometimes performs worse than having shared variables when no epoch prior is used. However, this is not the case on the harder CIFAR100 dataset or when the prior is used, justifying our choice of using the separate variables. Third, while the **FedSGD** method performs worse when the number of epochs $\mathcal{E}$ is large, all other methods, perhaps counter-intuitively, benefit from the additional epochs. We think this is due to the client images being used multiple times during the generation of the client update resulting in easier reconstructions. Finally, our experiments show that **FedSGD-Epoch** performs well when the number of batches per epoch $\mathcal{B}^c$ is small, but its performance degrades as more updates are made per epoch, to the point where it becomes much worse than FedSGD on batch size $m = 1$.

### 5.2 Label Count Reconstruction Experiments

We now experiment with the quality of our label count reconstruction method and compare it to prior work (Geng et al., 2021). We consider the following methods: (i) **Ours**, our label count reconstruction algorithm RecLabels described in Section 4.5, (ii) **Geng et al.** $\theta^s$, the label count reconstruction algorithm, as described in Geng et al. (2021), (iii) **Geng et al.** $\theta^c$, same as **Geng et al.** $\theta^s$, but with the parameters $\widetilde{O}^c$ and $\widetilde{p}_k^c$ estimated on the weights $\theta^c$ returned by the client.

In Table 2, we report the numbers of incorrectly reconstructed labels and their standard deviation for different values of the number of epochs $\mathcal{E}$, batch sizes $m$ and different methods averaged across users. Unlike image and text reconstructions where prior knowledge about the structure of inputs can be used to judge their quality, judging the quality of label count reconstructions is hard for an attacker as they do not have access to the client counts. To this end, a key property we require from label count reconstruction algorithms is that they are *robust* w.r.t. FedAvg parameters such as datasets, number of epochs $\mathcal{E}$ and batch sizes $m$, as choosing them on the fly is not possible without additional knowledge about the client label distribution. Results in Table 2 show that the algorithm in Geng et al. (2021), as originally proposed, performs well when the number of local FedAvg steps $U^c$ is small. Conversely, when approximating the parameters $\widetilde{O}^c$ and $\widetilde{p}_k^c$ at the client weights $\theta^c$, the algorithm works well predominantly when $U^c$ is large, except on CIFAR100 and $U^c = 500$.

Table 2: The effectiveness of our method and the baselines for the task of label counts reconstruction.

| Dataset | $\mathcal{E}$ | $m$ | $N^c$ | $U^c$ | Ours | Geng et al. $\theta^s$ | Geng et al. $\theta^c$ |
|---------|---|---|---|---|------|------|------|
| FEMNIST | 1 | 5 | 50 | 10 | $3.4 \pm 2.01$ | $\mathbf{3.2} \pm 1.84$ | $3.6 \pm 2.07$ |
| | 5 | 5 | 50 | 50 | $3.4 \pm 2.10$ | $7.8 \pm 6.77$ | $\mathbf{3.2} \pm 1.92$ |
| | 10 | 10 | 50 | 50 | $\mathbf{3.2} \pm 1.94$ | $8.2 \pm 7.06$ | $\mathbf{3.2} \pm 1.96$ |
| | 10 | 5 | 50 | 100 | $\mathbf{5.2} \pm 2.98$ | $9.2 \pm 4.35$ | $\mathbf{5.2} \pm 3.55$ |
| | 10 | 1 | 50 | 500 | $14.0 \pm 6.28$ | $15.4 \pm 4.73$ | $\mathbf{13.3} \pm 6.35$ |
| CIFAR100 | 1 | 5 | 50 | 10 | $4.2 \pm 1.41$ | $\mathbf{3.9} \pm 1.30$ | $4.6 \pm 1.55$ |
| | 5 | 5 | 50 | 50 | $3.2 \pm 1.24$ | $7.6 \pm 1.83$ | $\mathbf{2.5} \pm 1.09$ |
| | 10 | 10 | 50 | 50 | $2.8 \pm 1.22$ | $8.1 \pm 1.84$ | $\mathbf{2.0} \pm 0.86$ |
| | 10 | 5 | 50 | 100 | $\mathbf{4.9} \pm 1.57$ | $12.2 \pm 1.96$ | $5.2 \pm 1.45$ |
| | 10 | 1 | 50 | 500 | $\mathbf{8.1} \pm 1.77$ | $10.2 \pm 1.95$ | $15.7 \pm 3.48$ |

Table 3: The effectiveness of our method and the baselines for the image and label counts reconstruction.

| Dataset | $\mathcal{E}$ | $m$ | $N^c$ | $U^c$ | Ours (prior) | | Ours (no prior) | | Shared | | FedSGD-Epoch | | FedSGD | |
|---------|---|---|---|---|-------|------|-------|------|-------|------|-------|------|-------|------|
| | | | | | Rec(%) | PSNR | Rec(%) | PSNR | Rec(%) | PSNR | Rec(%) | PSNR | Rec(%) | PSNR |
| FEMNIST | 1 | 5 | 50 | 10 | **37.2** | **19.3** | **37.2** | **19.3** | **37.2** | **19.3** | 35.7 | 19.2 | 35.7 | 19.2 |
| | 5 | 5 | 50 | 50 | **51.9** | **20.3** | 47.5 | 20.1 | 40.8 | 19.5 | 47.0 | 19.8 | 12.5 | 17.3 |
| | 10 | 10 | 50 | 50 | **55.0** | **20.5** | 50.0 | 20.2 | 45.7 | 19.7 | 50.0 | 19.9 | 10.6 | 17.0 |
| | 10 | 5 | 50 | 100 | **48.5** | **20.2** | 43.0 | 19.8 | 43.6 | 19.3 | 45.8 | 19.1 | 7.2 | 16.6 |
| | 10 | 1 | 50 | 500 | **21.4** | **18.3** | 14.5 | 16.9 | 7.5 | 15.2 | 1.1 | 3.6 | 4.7 | 16.5 |
| CIFAR100 | 1 | 5 | 50 | 10 | **5.0** | **15.7** | **5.0** | **15.7** | **5.0** | **15.7** | 4.8 | **15.7** | 4.8 | **15.7** |
| | 5 | 5 | 50 | 50 | **46.7** | **18.5** | 45.0 | 18.3 | 7.0 | 15.8 | 7.7 | 15.8 | 0.8 | 14.1 |
| | 10 | 10 | 50 | 50 | **54.3** | **19.4** | 52.0 | 19.1 | 15.1 | 16.5 | 18.8 | 16.7 | 0.5 | 13.7 |
| | 10 | 5 | 50 | 100 | **46.0** | **18.5** | 43.2 | 18.2 | 8.7 | 15.7 | 8.1 | 14.9 | 0.6 | 13.1 |
| | 10 | 1 | 50 | 500 | **6.7** | **14.7** | 5.7 | 14.6 | 0.4 | 11.9 | 0.1 | 9.0 | 1.4 | 12.7 |

This makes it difficult to decide which variant of Geng et al. (2021) to apply in practice. In contrast, the results in Table 2 show that our label count reconstruction algorithm performs well across different settings. Further, it achieves the best reconstruction in the most challenging setting of CIFAR100 with small batch sizes ($m = 1, 5$) and its predictions often vary less than the alternatives. Therefore, we propose our method as the default choice for the FedAvg label count reconstruction algorithm due to its robustness.

### 5.3 End-to-End Attack

In this section, we show the results of our end-to-end attack, that combines our methods for label count and input reconstruction. We evaluate it under the same settings and compare it to the same methods as in Section 5.1. For all methods under consideration, we use our label count reconstruction method since, as we showed in Section 5.2, it performs well under most settings. We present the results in Table 3. Compared to the results in Table 1 (where we assumed label counts are known), the performance of our end-to-end attack remains strong even though we are now reconstructing the labels as well, showing the effectiveness of our label reconstruction. In particular, most results remain within 10% of the original reconstruction rate (resulting in $> 45\%$ success rate of our attack in most settings) except for the case of $U^c = 500$, where due to the higher error rate of the label reconstruction we observe more significant drop in performance. Note that even in this setting our attack performs the best and is able to reconstruct 20% of the FEMNIST images successfully. Furthermore, we observe that the **Shared** method is less robust to errors in the label counts compared to the other methods, as with known label counts it performs better than the **Ours (no prior)** method on FEMNIST, but the trend reverses when the label counts are imperfect. We think the reason is the sensitivity to label count errors of the shared variables optimization procedure which further justifies our use of separate optimization variables.

**Reconstruction Visualizations**  In addition to Table 3, we also provide visualizations of the reconstructions from the different attacks for $\mathcal{E} = 10$, $m = 5$, and $N^c = 50$ in Figure 2. For both FEMNIST and CIFAR, we show the reconstructions of 4 images from the first user's batch. We provide the full batch reconstructions in Appendix F.1. We observe that reconstructions obtained using our method are the least noisy. As we show in Appendix C.6, this is due to the matching and averaging effect. Further, we see that the prior has positive effect on the reconstruction as it can sharpen some of the edges in both the FEMNIST and CIFAR reconstructions. Finally, we also observe that the reconstructions of the **FedSGD** method are very poor.

Table 4: The effectiveness of our method with different priors on $10 \times 5 \times 50$ FEMNIST and CIFAR100.

| | | FEMNIST | | CIFAR100 | |
|---|---|---|---|---|---|
| $g$ | $D_{\mathrm{inv}}$ | Rec(%) | PSNR | Rec(%) | PSNR |
| mean | $\ell_1$ | 61.6 | 21.2 | 55.1 | 19.5 |
| conv + mean | $\ell_1$ | 61.5 | 21.2 | 56.0 | 19.6 |
| max | $\ell_1$ | 36.8 | 19.2 | 56.9 | 19.6 |
| conv + max | $\ell_1$ | 56.0 | 20.7 | 58.0 | **19.8** |
| mean | $\ell_2$ | **65.5** | **21.5** | 56.0 | 19.5 |
| conv + mean | $\ell_2$ | 64.4 | 21.4 | 56.3 | 19.6 |
| max | $\ell_2$ | 39.6 | 19.4 | 56.0 | 19.6 |
| conv + max | $\ell_2$ | 63.6 | 21.2 | **58.8** | **19.8** |

Figure 2: Reconstructions of 4 images from the $10 \times 5 \times 50$ FEMNIST and CIFAR end-to-end experiments.

## 5.4 Epoch Order-Invariant Priors

In this section, we investigate the effectiveness of various order-invariant functions $g$ and distance measures $D_{\mathrm{inv}}$. We experiment with the $\ell_1$ and $\ell_2$ distances for $D_{\mathrm{inv}}$, as they are the most natural choices for a distance functions. For $g$, inspired by PointNet (Qi et al., 2017) that also relies on the choice of an order-invariant function, we experiment with the mean and max functions. Further, following Li et al. (2022) that show image features produced by randomly initialized image networks can serve as a good image reconstruction priors, we also consider versions of $g$ that apply the mean and max functions on the result of applying 1 layer random convolution with 96 output channels, kernel size 3 and no stride on the images (denoted with conv + mean and conv + max in Table 4). As our performance increased by adding more output convolution channels, we chose the largest number of channels that fits in GPU memory. For both FEMNIST and CIFAR, we ran our experiments on $\mathcal{E} = 10$, $m = 5$, and $N^c = 50$ with known label counts. We chose $\lambda_{\mathrm{inv}}$ by exponential search on the $D_{\mathrm{inv}} = \ell_2$ and $g = \mathrm{mean}$ ($D_{\mathrm{inv}} = \ell_1$ and $g = \mathrm{conv} + \mathrm{max}$) combination on FEMNIST (CIFAR100). Due to the computational costs, we adapted these values of $\lambda_{\mathrm{inv}}$ to the other combinations in a way that the value of $\lambda_{\mathrm{inv}} \cdot \mathcal{L}_{\mathrm{inv}}(\widetilde{X}^c)$ matches on average at the first iteration of the optimization process. Comparison is performed on the same 100 clients as before, as the attacker can select the prior which obtains the highest quality of the reconstruction. The results are shown in Table 4. We observe that $D_{\mathrm{inv}} = \ell_2$ performs consistently better in all experiments and thus we chose to use it. Furthermore, on FEMNIST $g$ does not benefit from the additional complexity of the convolution, likely due to being simpler dataset and that the max is a very poor choice for this dataset. To this end, we choose $g$ to be simply the mean. In contrast, on CIFAR we observe that the random convolution is always helpful to the reconstruction. Also, we see that the combination of the convolution and the max is particularly effective. To this end, we choose to use this combination on CIFAR. We believe more complex feature extracting functions such as ones that use stacks of convolutions or more output channels will have even greater benefit on this dataset. We leave that as future work.

## 6 Conclusion

In this work, we presented a new data leakage attack for the commonly used federated averaging learning algorithm. The core ingredients of our attack were a novel simulation-based reconstruction loss combined with an epoch order-invariant prior, and our FedAvg-specific extension of existing label reconstruction algorithms. We experimentally showed that our method can effectively attack FedAvg updates computed on combinations of large number of epochs and batches per epoch, thus for the first time demonstrating that realistic FedAvg updates are vulnerable to data leakage attacks. We believe that our results indicate a need for a more thorough investigation of data leakage in FedAvg and refer to our Broader Impact Statement in Appendix E, that further discusses the practical implications of our work.

**Acknowledgments**

This publication was made possible by an ETH AI Center postdoctoral fellowship granted to Nikola Konstantinov.

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

---

**Algorithm 3** Overview of our matching and averaging algorithm

---

1:  **function** MatchEpoch($\widetilde{X}^c$)
2:      $\widetilde{X}^c \leftarrow$ ReorderEpoch($\widetilde{X}^c$)
3:      **for** $i \leftarrow 1, \ldots, N^c$ **do**
4:          $b_i \leftarrow \lfloor \frac{i}{m} \rfloor$
5:          $\mathcal{I} \leftarrow i - b_i \cdot m$
6:          $\widetilde{X}_i^c \leftarrow \frac{1}{\mathcal{E}} \sum_{e=1}^{\mathcal{E}} \widetilde{X}_{e,b_i,\mathcal{I}}^c$
7:      **end for**
8:      **return** $\{\widetilde{X}_i^c \mid i \in [N^c]\}$
9:  **end function**

10: **function** ReorderEpoch($\widetilde{X}^c$)
11:     **for** $e \in 2, \ldots, \mathcal{E}$ **do**
12:         **for** $(i, j) \in [N^c] \times [N^c]$ **do**
13:             $b_i \leftarrow \lfloor \frac{i}{m} \rfloor$
14:             $b_j \leftarrow \lfloor \frac{j}{m} \rfloor$
15:             $\mathcal{I} \leftarrow i - b_i \cdot m$
16:             $\mathcal{J} \leftarrow j - b_j \cdot m$
17:             $M_{i,j} = \text{sim}(\widetilde{X}_{1,b_i,\mathcal{I}}^c, \widetilde{X}_{e,b_j,\mathcal{J}}^c)$
18:         **end for**
19:         $\text{order}_e \leftarrow$ LinSumAssign($M$)
20:         $\{\widetilde{X}_{e,b}^c \mid b \in [\mathcal{B}^c]\} \leftarrow$ Reorder($\{\widetilde{X}_{e,b}^c \mid b \in [\mathcal{B}^c]\}, \text{order}_e$)
21:     **end for**
22:     **return** $\widetilde{X}^c$
23: **end function**

---

# A  Matching and Averaging Algorithm

In this section, we present in details our matching and averaging method that takes our per-epoch reconstructions $\widetilde{X}^c$ and combines them into our final input reconstructions $\{\widetilde{X}_i^c \mid i \in [N^c]\}$. The method is shown in Algorithm 3. The matching part of the method is separated in the function ReorderEpoch which computes an optimal reordering of the reconstructions at every epoch $e$ so that its reconstructions match the ones in the first epoch best. To this end, we compute a similarity measure between every image $\widetilde{X}_{1,b_i,\mathcal{I}}^c$ in every batch $b_i$ in the first epoch and every image $\widetilde{X}_{e,b_j,\mathcal{J}}^c$ in every batch $b_j$ in epoch $e$, for every epoch $e$ except the first one, and store them in the matrix $M$ (Line 12–18). For the image experiments in this paper, we use the PSNR similarity measure. Once $M$ is computed, we use the linear sum assignment problem solver LinAssign provided by SciPy (Virtanen et al., 2020) to find the optimal reordering $\text{order}_e$ of the images in epoch $e$ (Line 19–20). Finally, the matched inputs are then averaged across epochs in the MatchEpoch function (Line 3–7).

# B  Further Experimental Details

## B.1  Network Details

In Table 5, we show the neural network architectures we use for our FEMNIST and CIFAR100 experiments. They both consists of 2 convolutional layers with average pooling, followed by 2 linear layers. As CIFAR is a harder dataset than FEMNIST, we double the sizes of the layers in the CIFAR architecture.

## B.2  Attack Hyperparameters

In all experiments, we use total variation (TV) (Estrela et al., 2016) as an additional regularizer, similarly to Geiping et al. (2020), as well as, the clipping regularizer $\mathcal{R}_{\text{clip}}$ presented in Equation 17 in Geng et al.

Table 5: The network architectures for the networks attacked in this paper.

| Conv2d(in_channels=3, out_channels=32, kernel_size=3, stride=1, padding=1) |
| --- |
| ReLU() |
| AvgPool2d(kernel_size=2, stride=2) |
| Conv2d(in_channels=32, out_channels=64, kernel_size=1, padding=1) |
| ReLU() |
| AvgPool2d(kernel_size=2, stride=2) |
| Linear(in_features=5184, out_features=100) |
| ReLU() |
| Linear(in_features=100, out_features=62) |

(a) FEMNIST network architecture.

| Conv2d(in_channels=3, out_channels=64, kernel_size=3, stride=1, padding=1) |
| --- |
| ReLU() |
| AvgPool2d(kernel_size=2, stride=2) |
| Conv2d(in_channels=64, out_channels=128, kernel_size=1, padding=1) |
| ReLU() |
| AvgPool2d(kernel_size=2, stride=2) |
| Linear(in_features=10368, out_features=200) |
| ReLU() |
| Linear(in_features=200, out_features=100) |

(b) CIFAR100 network architecture.

(2021). We balance them with the rest of our reconstruction loss $\ell$ using the hyperparameters $\lambda_{\text{TV}}$ and $\lambda_{\text{clip}}$, respectively. Additionally, we use different learning rates $\eta_{\text{rec}}$ and learning rate decay factors $\gamma_{\text{rec}}$ for solving the optimization problem in Algorithm 2. For both datasets we use 200 optimization steps and client learning rate $\eta = 0.004$ which was suggested as good learning rate for FEMNIST in LEAF(Caldas et al., 2018). We selected the rest of the hyperparameters by optimizing the performance of the FedSGD reconstruction on FedSGD updates with 50 images. We did this on FEMNIST and CIFAR100 separately. The resulting hyperparameters are shown below and used for all experiments.

**FEMNIST** We use the following hyperparameters for our FEMNIST experiments $\lambda_{\text{TV}} = 0.001$, $\lambda_{\text{clip}} = 2$, $\lambda_{\text{inv}} = 1000$, $\eta_{\text{rec}} = 0.4$, and exponential learning rate decay $\gamma_{\text{rec}} = 0.995$ applied every 10 steps.

**CIFAR100** We use the following hyperparameters for our CIFAR100 experiments $\lambda_{\text{TV}} = 0.0002$, $\lambda_{\text{clip}} = 10$, $\lambda_{\text{inv}} = 6.075$, $\eta_{\text{rec}} = 0.1$, and exponential learning rate decay $\gamma_{\text{rec}} = 0.997$ applied every 20 steps.

## C  Further Experiments

In this section we present a number of additional experiments that further validate the effectiveness of our attack. In all cases, we focus for simplicity on the FEMNIST dataset and we use $\mathcal{E} = 10$ epochs and batch size $m = 5$ on $N^c = 50$ images. Unless otherwise stated, we assume known label counts per epoch as in Section 5.1.

### C.1  Attacking Defended Networks

First, we experiment with attacking FL protocols that adopt known defenses against data leakage. In particular, we consider defending by adding small amounts of Gaussian and Laplacian noise to the communicated gradients. We note that if clipping is additionally applied this is equivalent to the differential privacy defenses in Abadi et al. (2016). Additionally, we experiment with a defense that randomly prunes the entries of the gradient with a certain probability, as suggested by Zhao et al. (2020). For each of the defense methods, we present the results of the attacks in Section 5.1 at two different defense strength levels in Table 6. We chose the defense strengths such that the resulting networks lose around 1% and 4% accuracy, respectively. We also report the resulting network test-set accuracies after training for 250 communication rounds, each using 10 random clients.

As expected, the results show that stronger defenses sacrifice more network accuracy in exchange for more data privacy. Even so, our attack is able to reconstruct $> 15\%$ of the images on all strongly defended networks while also being the best performing method across the board. Further, we see that the noise-based defenses provide a better trade-off between accuracy and defense performance. This is expected, due to their connection with differential privacy. In particular, we see that the Gaussian noise defense results in only 17.2% reconstructed images, while achieving 66.14% accuracy, which results in the best trade-off among the methods we considered.

Table 6: The effectiveness of the variations of our method and the baselines with known labels and different defenses on FEMNIST with 10 epochs of batch size 5 and 50 images.

| | | Ours (prior) | | Ours (no prior) | | Shared | | FedSGD-Epoch | | FedSGD | |
|---|---|---|---|---|---|---|---|---|---|---|---|
| Defense | Accuracy(%) | Rec(%) | PSNR | Rec(%) | PSNR | Rec(%) | PSNR | Rec(%) | PSNR | Rec(%) | PSNR |
| No Defense | 70.09 | **65.5** | **21.5** | 50.6 | 20.3 | 59.9 | 20.3 | 56.0 | 19.6 | 8.3 | 16.7 |
| Gaussian Noise, $\sigma = 0.01$ | 68.91 | **48.0** | **20.2** | 36.8 | 19.1 | 25.0 | 18.3 | 24.9 | 17.9 | 7.5 | 16.6 |
| Gaussian Noise, $\sigma = 0.03$ | 66.14 | **17.2** | **16.8** | 13.1 | 15.5 | 2.8 | 13.0 | 4.5 | 13.2 | 4.5 | 15.6 |
| Laplacian Noise, $\sigma = 0.01$ | 68.88 | **38.9** | **19.4** | 29.3 | 18.4 | 14.9 | 17.1 | 15.6 | 16.7 | 6.6 | 16.4 |
| Laplacian Noise, $\sigma = 0.02$ | 65.94 | **19.0** | **17.0** | 14.4 | 15.7 | 3.7 | 13.4 | 5.0 | 13.5 | 4.8 | 15.7 |
| Random Pruning, $p = 0.2$ | 68.84 | **51.5** | **20.4** | 40.2 | 19.5 | 40.5 | 19.2 | 40.8 | 18.9 | 9.2 | 17.0 |
| Random Pruning, $p = 0.5$ | 66.62 | **27.6** | **18.7** | 22.3 | 18.1 | 18.3 | 17.5 | 18.6 | 17.5 | 10.2 | 17.2 |

Table 7: The effectiveness of the variations of our method and the baselines with known labels at different training rounds on FEMNIST with 10 epochs of batch size 5 and 50 images.

| | Ours (prior) | | Ours (no prior) | | Shared | | FedSGD-Epoch | | FedSGD | |
|---|---|---|---|---|---|---|---|---|---|---|
| Round | Rec(%) | PSNR | Rec(%) | PSNR | Rec(%) | PSNR | Rec(%) | PSNR | Rec(%) | PSNR |
| 1 | **65.5** | **21.5** | 50.6 | 20.3 | 59.9 | 20.3 | 56.0 | 19.6 | 8.3 | 16.7 |
| 6 | **58.8** | **20.8** | 27.4 | 16.2 | 39.2 | 17.9 | 22.8 | 15.6 | 7.3 | 15.4 |
| 11 | **38.4** | **18.0** | 7.4 | 13.1 | 21.3 | 15.4 | 10.8 | 13.2 | 5.9 | 15.4 |

## C.2 Trained Networks Experiments

Next, we present our experiments in which the attack is conducted further into the training process, rather than during the first communication round. In Table 7, we report the results for all attack methods from Section 5.1 when applied at FedAvg communication rounds 6 and 11 and compare them to the results from the main body (Round=1). We use 10 clients per communication round. While the effectiveness of the attacks is reduced further into the training, our method is still able to recover as much as 38.4% of the images at the later stages of the optimization process. In addition, the benefits of our design choices, in particular using the order-invariance prior and epoch-specific optimization variables, become more apparent in this context, as the performance of the rest of the attack methods in Table 7 is significantly reduced compared to our full attack.

## C.3 Label Order Experiments

Additionally, we conduct two ablations studies to evaluate the effectiveness of our attacks depending on whether the split into batches at each local epoch is random or consistent across epochs and on whether the server has knowledge of the label counts per batch or not. We present the results of this ablation study in Table 8. We observe that while the addition of the batch label count information results in consistently better reconstructions, for most methods the gap is not big. This suggests that our choice of randomly assigning the labels into batches works well in practice. The exception is the **Shared** method that really benefits from the additional information and with it, it is capable of matching the performance of our method in terms of number of reconstructed images even though it still achieves lower PSNR. We believe this is due to the sensitivity of the optimization problem when shared variables are used and motivates our choice of keeping separate optimization variables per epoch when the label counts per batch is unknown. Another somewhat surprising observation in Table 8 is that using random batches actually makes the reconstruction problem slightly easier. We hypothesize that the randomization helps as every image gets batched with different images at every epoch, making it easier to disambiguate it from the other images in the batch in at least one of the epochs. This suggests that using consistent batches might be a cheap way to increase the privacy of FedAvg updates.

Table 8: The effectiveness of the variations of our method and the baselines with known labels depending on the information available to the server on FEMNIST with 10 epochs of batch size 5 and 50 images.

| Known Counts per Batch | Random Batches | Ours (prior) | | Ours (no prior) | | Shared | | FedSGD-Epoch | | FedSGD | |
|---|---|---|---|---|---|---|---|---|---|---|---|
| | | Rec(%) | PSNR | Rec(%) | PSNR | Rec(%) | PSNR | Rec(%) | PSNR | Rec(%) | PSNR |
| ✓ | ✓ | 68.2 | **21.7** | 52.8 | 20.4 | **68.4** | 21.1 | 56.0 | 19.6 | 8.3 | 16.7 |
| ✓ | ✗ | **63.4** | **21.3** | 49.4 | 20.1 | **63.4** | 20.6 | 52.7 | 19.4 | 7.9 | 16.7 |
| ✗ | ✓ | **65.5** | **21.5** | 50.6 | 20.3 | 59.9 | 20.3 | 56.0 | 19.6 | 8.3 | 16.7 |
| ✗ | ✗ | **60.7** | **21.1** | 46.6 | 19.9 | 55.4 | 20.0 | 52.7 | 19.4 | 7.9 | 16.7 |

Table 9: The effectiveness of the variations of our method and the baselines with known labels and known values of $N^c$ and $U^c$ but unknown values of $\mathcal{E}$ and $m$. The True columns represent the value used by the clients to compute their FedAvg update, while the Chosen columns represent the values used by the algorithms during reconstruction.

| $\mathcal{E}$ | | $m$ | | $N^c$ | | $U^c$ | | Ours (prior) | | Ours (no prior) | | Shared | | FedSGD-Epoch | | FedSGD | |
|---|---|---|---|---|---|---|---|---|---|---|---|---|---|---|---|---|---|
| True | Chosen | True | Chosen | True | Chosen | True | Chosen | Rec(%) | PSNR | Rec(%) | PSNR | Rec(%) | PSNR | Rec(%) | PSNR | Rec(%) | PSNR |
| 5 | 5 | 5 | 5 | 50 | 50 | 50 | 50 | **63.1** | **21.2** | 56.2 | 20.7 | 59.0 | 20.6 | 56.1 | 20.4 | 13.7 | 17.4 |
| 5 | 10 | 5 | 10 | 50 | 50 | 50 | 50 | **55.8** | **20.7** | 55.2 | 20.6 | 53.8 | 20.1 | 53.8 | 20.1 | 12.7 | 17.3 |
| 5 | 25 | 5 | 25 | 50 | 50 | 50 | 50 | 51.0 | 20.2 | **51.7** | **20.3** | 40.5 | 19.0 | 39.7 | 19.0 | 11.4 | 17.1 |
| 5 | 50 | 5 | 50 | 50 | 50 | 50 | 50 | **52.0** | **20.2** | 51.4 | 20.2 | 34.3 | 18.5 | 34.6 | 18.5 | 10.8 | 17.0 |

## C.4 Data Reconstruction without Knowledge of $\mathcal{E}$ and $m$

In the paper, so far, we assumed that the server has knowledge of $\mathcal{E}$, $m$, and $N^c$ when mounting the reconstruction attack. This allows the server to further calculate $\mathcal{B}^c$ and $U^c$, as well, giving it full knowledge over the clients' dataset parameters. In this section, we look at a weaker attack model where the server only knows $U^c$ and $N^c$. This attack model is motivated by the fact that the FedAvg server only needs $U^c$ and $N^c$ to decide how to weight the different client updates when aggregating them, suggesting that the clients can choose not to share $\mathcal{E}$ and $m$. One possible way to mitigate this, is to run our reconstruction with multiple values of $\mathcal{E}$ and $m$ and pick the best reconstruction. To this end, we test the robustness of our method to using a wrong value of $\mathcal{E}$ in our reconstruction. In particular, we generate FedAvg client updates with $\mathcal{E} = 5$, $m = 5$ and $N^c = 50$ on the FEMNIST dataset and then reconstruct data from them by plugging different values of $\mathcal{E}$ and $m$ in our algorithm. As $U^c = \mathcal{E} \cdot \mathcal{B}^c$ is avaliable to the attacker, it is known that $\mathcal{E}$ is a divisor of $U^c = 50$. Thus, we choose $\mathcal{E}$ to be either 10, 25 or 50, resulting in values for $\mathcal{B}^c$ of 5, 2 and 1, respectively. In this experiment, we assume that all batches in the client are of equal size, thus allowing us to compute $m$ using the formula $m = \frac{N^c}{\mathcal{B}^c}$. The results of reconstructing the data with these (wrong) choices of the parameters $\mathcal{E}$ and $m$ for the methods in Table 1 and known labels are presented in Table 9. In Table 9, we also present the reconstruction with the correct values of $\mathcal{E}$ and $m$ for comparison. We make several observations. First, the image reconstruction becomes worse the further the chosen value of $\mathcal{E}$ is from the correct value, 5, for all methods. Despite this, our method still outperforms the baselines and it is capable of reconstructing more than 50% of the images in all cases. Second, our order-invariant prior is less effective when the true value of $\mathcal{E}$ is unknown. We think there are two reasons for that. One is that we rely on the value chosen for $\lambda_{\text{inv}}$ originally on the correct values of $\mathcal{E}$ and $m$ in all experiments in Table 9 which can be suboptimal when $\mathcal{E}$ is chosen wrongly. The other is that our prior relies on the knowledge of the correct number of epochs $\mathcal{E}$ and therefore is less effective when the wrong value of $\mathcal{E}$ is used. Finally, unlike our method, we see that the performance of the shared variable method degrades drastically when the choice of $\mathcal{E}$ is wrong. This is in line with our other experiments where we also see this method is less robust with respect to changes to the FedAvg client updates.

## C.5 Reconstructing Data from Aggregated Updates

In this section, we experiment with reconstructing images from updates aggregated between several clients. The experiment was carried out on the FEMNIST dataset with individual client updates consisting of 10 epochs with 10 batches of 5 images each. The aggregated updates were computed by taking the mean of the client updates participating in the aggregation. We show the results in Table 10, where the first column

Table 10: The effectiveness of the variations of our method and the baselines for reconstructing data from aggregated updates with known labels on FEMNIST with 10 epochs of batch size 5 and 50 images.

| # clients | Ours (prior) | | Ours (no prior) | | Shared | | FedSGD-Epoch | | FedSGD | |
|---|---|---|---|---|---|---|---|---|---|---|
| | Rec(%) | PSNR | Rec(%) | PSNR | Rec(%) | PSNR | Rec(%) | PSNR | Rec(%) | PSNR |
| 1 | **65.5** | **21.5** | 50.6 | 20.3 | 59.9 | 20.3 | 56.0 | 19.6 | 8.3 | 16.7 |
| 2 | **42.1** | **19.8** | 32.3 | 18.7 | 27.6 | 18.3 | 25.7 | 17.7 | 6.8 | 16.1 |
| 4 | **29.7** | **18.8** | 24.8 | 17.8 | 14.8 | 17.0 | 14.3 | 16.3 | 5.7 | 15.5 |

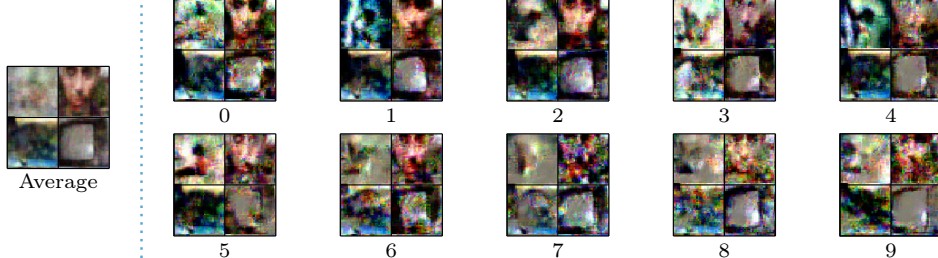

Figure 3: An end-to-end reconstruction of 4 images at individual epochs from the $10 \times 5 \times 50$ CIFAR100.

represents the number of participating clients. Thus, the first row in Table 10, showing the results of using only 1 client, corresponds to the experiment presented in Table 1. Similarly to Table 1, we report the average reconstruction results across 100 different aggregated updates and we assume that the total label counts for each client is known, but the counts per batch are not. To account for the aggregated updates, we changed SIMUPDATE to simulate all client updates separately with the procedure described in Section 4.3 and then average them to produce the final simulated aggregated update used in Algorithm 2. From the results, we see that as more client updates are aggregated we obtain lower reconstruction rates for all of the attacks. However, our attack method reconstructs the most images in all settings compared to the alternatives, while achieving higher PSNR. Further, our method is capable of reconstructing $\approx 30\%$ of the images when updates are aggregated across 4 different clients, suggesting that aggregated updates are vulnerable when aggregation with a small number of clients is used.

### C.6   Importance of averaging across epochs

In this section we visualize the effect of the matching and averaging, described in Section 4.4, used to generate the final reconstructions for our method. In Figure 3, we show the reconstructions at different epochs for our end-to-end attack computed on the same setup as Section 5.4 with $\mathcal{E} = 10$, $m = 5$, and $N^c = 50$. Further, we provide full batch reconstructions with known and unknown label counts in Appendix F.2. We make two observations. First, the reconstructions at later epochs are of worse quality. We think this is due to the imperfect label reconstruction making the simulation worse at later epochs, since in Appendix F.2 we show this effect is not seen when the label counts are known. Second, while the reconstructions at individual epochs are noisy, the average reconstructs the original images significantly better than any individual epoch reconstruction. This confirms that matching and averaging substantially improves the reconstruction results.

## D   Computational Complexity

### D.1   Runtime and Memory Complexity of SIMUPDATE

The forward pass through SIMUPDATE has exactly the same time complexity as a regular FedAvg update. Thus, it has $O(U^c \cdot T_{forw+back})$ time complexity, where $U^c$ is the number of local weight update steps used to compute the original FedAvg update and $T_{forw+back}$ is the time it takes to compute a forward and backward pass through the network on a single batch of data. Further, SIMUPDATE's forward pass has $O(N^c)$ memory

Table 11: Runtimes of our method and the baselines on NVIDIA RTX 2080 Ti GPU for the FEMNIST experiments originally presented in Table 1, measured in hours.

| Dataset | $\mathcal{E}$ | $m$ | $N^c$ | $U^c$ | Ours (prior) | Ours (no prior) | Shared | FedSGD-Epoch | FedSGD |
|---|---|---|---|---|---|---|---|---|---|
| | 1 | 5 | 50 | 10 | 0.50 | 0.50 | 0.50 | **0.25** | **0.25** |
| | 5 | 5 | 50 | 50 | 2.25 | 2.25 | 2.00 | 0.75 | **0.25** |
| FEMNIST | 10 | 10 | 50 | 50 | 2.50 | 2.50 | 2.50 | 1.25 | **0.25** |
| | 10 | 5 | 50 | 100 | 4.25 | 4.25 | 4.50 | 1.50 | **0.25** |
| | 10 | 1 | 50 | 500 | 7.50 | 7.50 | 7.50 | 1.25 | **0.50** |
| | 1 | 5 | 50 | 10 | 3.50 | 3.50 | 3.50 | **3.00** | **3.00** |
| | 5 | 5 | 50 | 50 | 18.00 | 17.00 | 17.00 | 14.50 | **3.00** |
| CIFAR100 | 10 | 10 | 50 | 50 | 35.50 | 30.50 | 35.00 | 28.50 | **3.00** |
| | 10 | 5 | 50 | 100 | 36.00 | 35.50 | 33.50 | 29.00 | **3.00** |
| | 10 | 1 | 50 | 500 | 7.00 | 5.50 | 5.50 | 29.00 | **3.50** |

complexity in the case of shared epoch variables as it requires to fit all client images in GPU memory and $O(\mathcal{E} \cdot N^c)$ in the case of individual epoch variables.

### D.2 Runtime Comparison

In Table 11, we provide a comparison between the total runtime of the different variants of our method and the baselines for the FEMNIST experiments originally presented in Table 1. Predictably, methods based on simulation, while obtaining significantly better reconstructions (See Table 1), are several times slower to execute than FedSGD. On FEMNIST, the FedSGD-Epoch method presents an interesting tradeoff between reconstruction results and speed as its runtime is much closer to the FedSGD method while still being able to reconstruct a big portion of the images in many settings. It is important to note that in practice the attacker can store the client's updates locally and only later mount the attack outside of the FedAvg learning process. Thus, the time taken to reconstruct the client data is less important than obtaining the data accurately, provided that the attack itself can be executed in a reasonable time.

## E    Broader Impact Statement

This paper proposes an algorithm for data leakage in federated learning, that is, a method for reconstructing clients' data from their gradient updates only. Given that our attack is applicable to the general FedAvg setting, it could be used by real-world FL service providers for reconstructing private data.

We believe that reporting the existence of effective data leakage methods like ours is a crucial step towards making future FL protocols less vulnerable to such attacks. Indeed, understanding the vulnerabilities of currently used algorithms can inspire further work on defending private data from malicious servers and on understanding what FL algorithms provide optimal protection in such scenarios. In addition, our evaluation in Appendix C.1 suggest that existing techniques that add noise to gradients and but come with some reduction of model accuracy, might be necessary in order to protect the data of the participating clients.

## F    Additional Visualizations

In this section, we present the complete batch visualizations of the images reconstructed by our method and baselines, a limited version of which were presented in the main text.

### F.1 Reconstruction Visualizations

First we visualize the reconstructions of all the data of the first user on the FEMNIST and CIFAR100 experiment with 10 epochs, batch size 5 and 50 images per client. Figures Figure 4 and Figure 6 show the results when the labels are available to the server, while Figure 5 and Figure 7 present the reconstructions for when the labels are unknown (similarly to the visualizations in the main body).

We see that our attack is able to reconstruct many of the images with significant accuracy while being less noisy than the other methods. We also note that, naturally, prior knowledge of the labels counts helps for the image reconstruction and in particular the recovered images are less blurry.

### F.2 Epoch Reconstruction Visualizations

We also provide the complete batch visualizations for the experiment in Figure 3 from the main body for both FEMNIST and CIFAR100. Results for when the epoch label counts are known are shown in Figure 8 and Figure 10, while those for unknown label counts are presented in Figure 9 and Figure 11.

We observe that, overall, our reconstruction performs better during the earlier local epochs, similarly to the observations in the main text. However, this effect is less expressed for the case when the labels are known, possibly because the resulting simulation is more accurate.

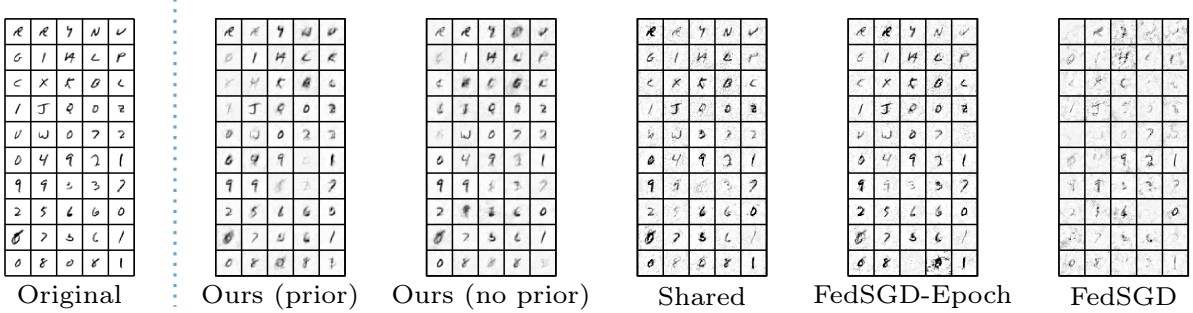

Figure 4: A reconstruction of the images from the $10 \times 5 \times 50$ FEMNIST experiment with known labels on the variations of our method and the baselines.

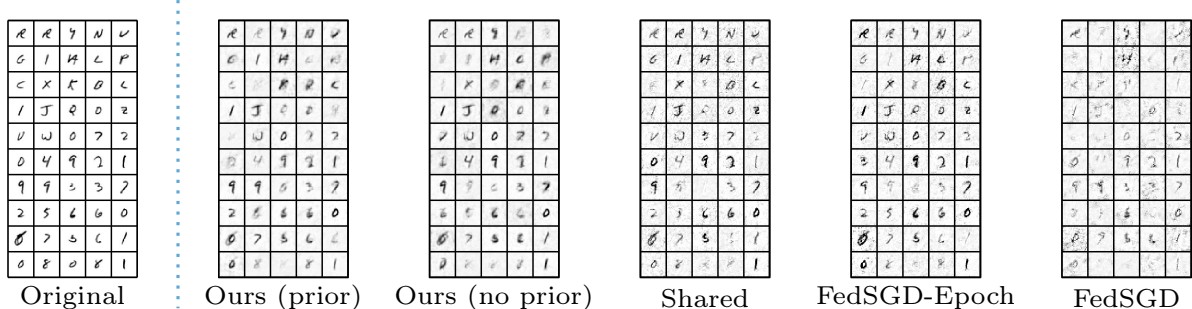

Figure 5: A reconstruction of the images from the $10 \times 5 \times 50$ FEMNIST experiment with label reconstruction on the variations of our method and the baselines.

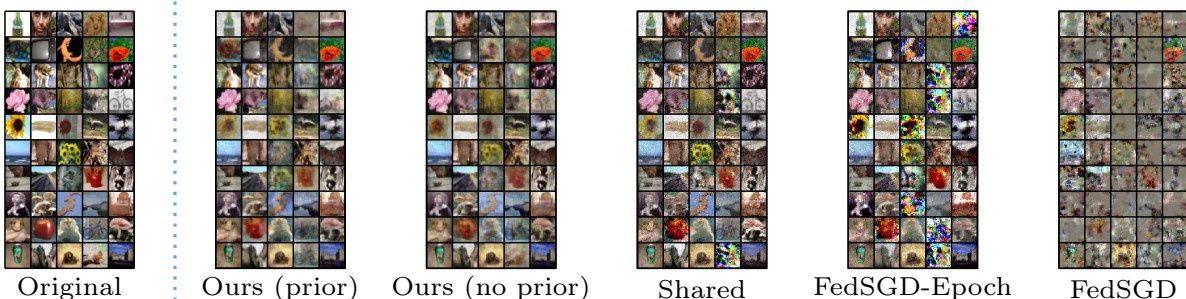

Figure 6: A reconstruction of the images from the $10 \times 5 \times 50$ CIFAR100 experiment with known labels on the variations of our method and the baselines.

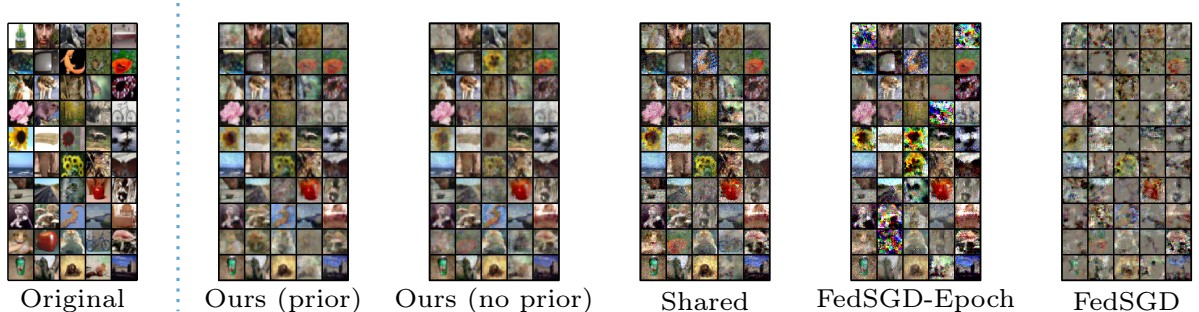

Figure 7: A reconstruction of the images from the $10 \times 5 \times 50$ CIFAR100 experiment with label reconstruction on the variations of our method and the baselines.

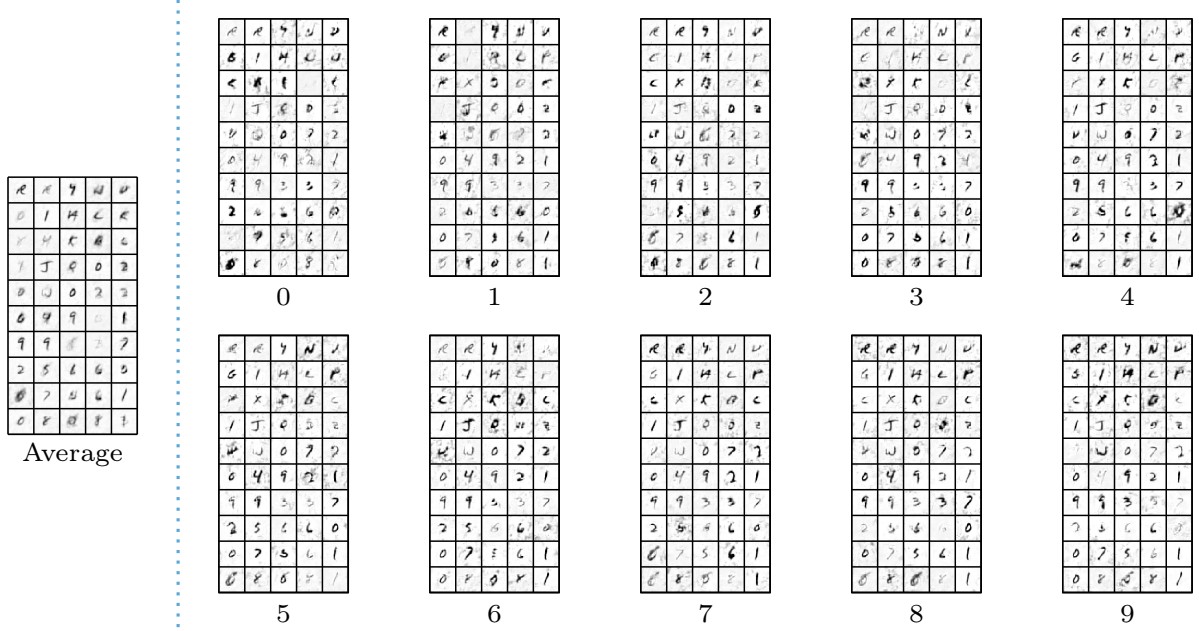

Figure 8: A reconstruction of the images at individual epochs from the $10 \times 5 \times 50$ FEMNIST experiment with known labels by our full method.

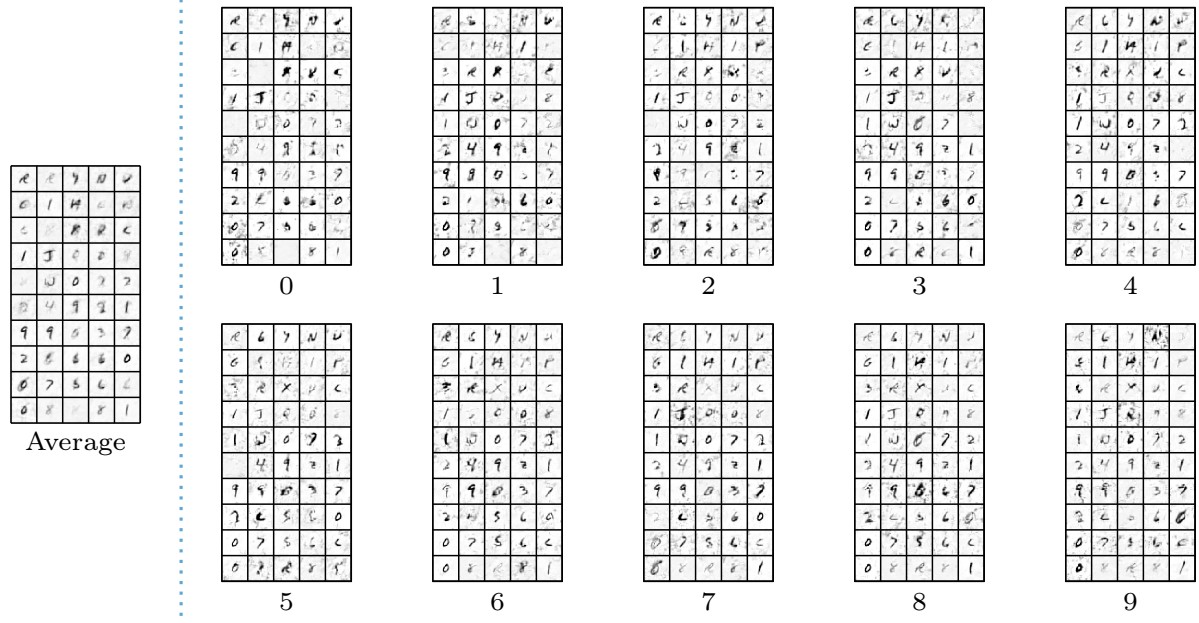

Figure 9: A reconstruction of the images at individual epochs from the $10 \times 5 \times 50$ FEMNIST experiment with label reconstruction by our full method.

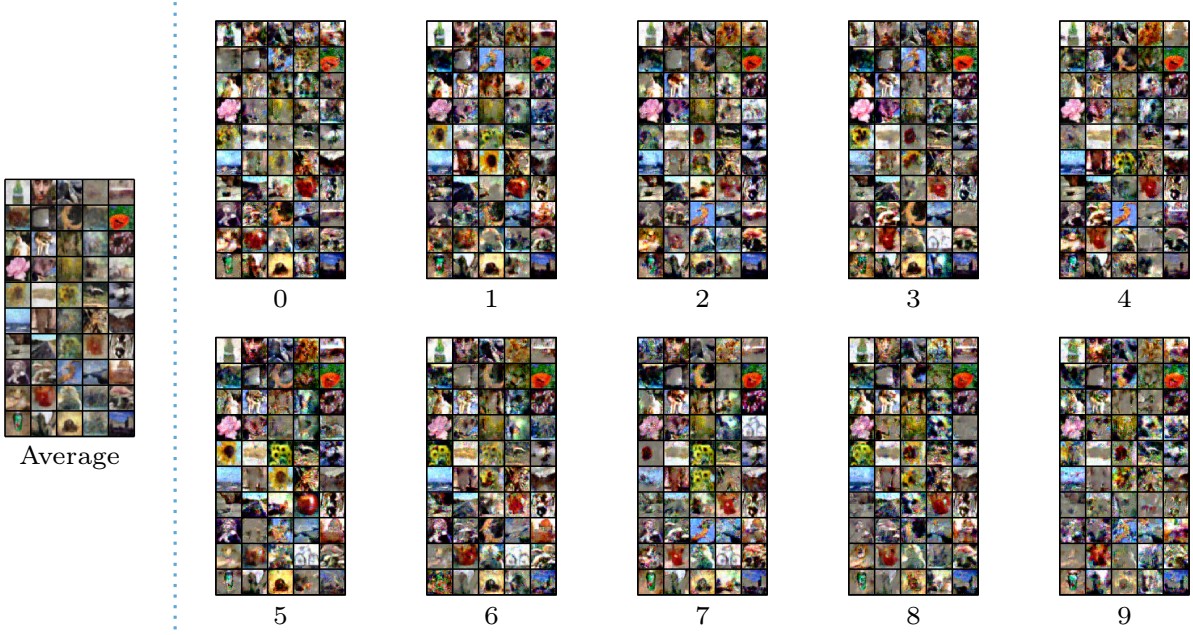

Figure 10: A reconstruction of the images at individual epochs from the $10 \times 5 \times 50$ CIFAR100 experiment with known labels by our full method.

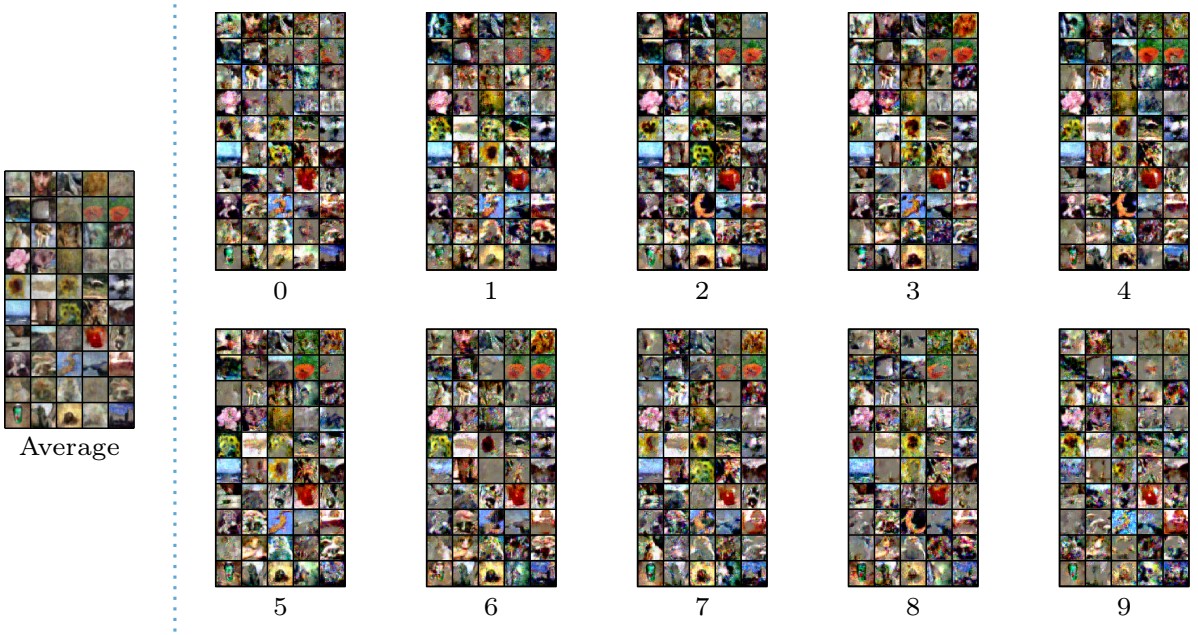

Figure 11: A reconstruction of the images at individual epochs from the $10 \times 5 \times 50$ CIFAR100 experiment with label reconstruction by our full method.

