# OpenReview forum: "Data Leakage in Federated Averaging"
_TMLR — Accepted by TMLR_

### Review · Reviewer_EebW · 2022-07-08

**Summary Of Contributions:**

This paper proposed an algorithm to reconstruct images from model delta (aggregated gradient updates) as an attack for federated learning. The main contribution is that the attack considers model deltas from multiple local steps instead of a single gradient. The proposed method is evaluated on EMNIST and CIFAR-100 datasets.

**Broader Impact Concerns:**

I would encourage the authors discuss limitations and possible defenses of the proposed method.

**Requested Changes:**

I feel the general writing/organization can be significantly improved. For example,

 - In section 4.3, how to get \tilt{X}? What is the complexity of “SimUpdate”, i.e., reconstruction time and resources? What is the exact reconstruction loss?

- In section 4.5 and section 5, what is U^{c}, and why would it affect the performance in table 1 and 3.

- In section 5, how are “the percentage of images that were successfully reconstructed by the attack” computed?


If I understand correctly, the reconstruction assumes the model updates from each client can be observed. What if only aggregated results from multiple clients (e.g., Federated Learning and Privacy https://dl.acm.org/doi/pdf/10.1145/3500240, https://arxiv.org/abs/2202.00580) can be observed?





**Strengths And Weaknesses:**

Strengths
+ The paper aims to evaluate an important problem. The FedAvg algorithm is generally more popular than FedSGD. The local steps for communication efficiency is one of the main techniques in FL. Proposing a more practical attack that can work with multiple local steps can help us audit and develop privacy principles.

Weakness
- The main weakness is the clarity of writing. I am not confident that I understand the algorithm to reproduce the results.
- I would hope to see more discussions on the attack and defense practice. Specifically, [Huang et al. 2021 Evaluating Gradient Inversion Attacks and Defenses in Federated Learning] suggests that attacks often make unrealistic assumptions and very simple defense methods can have significant impact.

---

> ### Author Response · Authors · 2022-07-13
> **Response to Reviewer EebW Part 1/2**
>
> We thank the reviewer for his comments and corroboration of the importance of the problem being tackled in the paper. We will revise the paper to address the quality of writing concerns once all reviews have become available, as per TMLR’s guidelines. In the meantime, we provide clarifications and answer your questions below:
>
> **Q:** In section 4.3, how to get $\widetilde{X}^c$?
>
> **A:** $\widetilde{X}^c$ are the algorithm’s optimization variables. As such, they are randomly initialized once (Line 5 in Algorithm 2) and then updated at each optimization step (Line 10 in Algorithm 2), so that their simulated update matches the real data $X^c$. In Section 4.3, we give two different possible structures for $\widetilde{X}^c$ depending on whether SimUpdate uses shared or separate optimization variables per epoch. That is if the same variable is used to compute the update on Line 8, Algorithm 1 across epochs or not. The structure is decided at initialization time (Line 5 in Algorithm 2) depending on the version of the algorithm we choose to run.
>
> **Q:** What is the complexity of “SimUpdate”, i.e., reconstruction time and resources?
>
> **A:** The forward pass through SimUpdate has exactly the same time complexity as a regular FedAvg update. Thus it has $O(U^c \cdot T_{forw+back})$ time complexity, where $U^c$ is the number of local weight update steps used to compute the original FedAvg update and $T_{forw+back}$ is the time it takes to compute a forward and backward pass through the network on a single batch of data. Further, SimUpdate’s forward pass has $O(N^c)$ memory complexity in the case of shared epoch variables as it requires to fit all client images in GPU memory and $O(\mathcal{E} \cdot N^c)$ in the case of individual epoch variables.
>
> **Q:** What is the exact reconstruction loss in Section 4.3?
>
> **A:** The reconstruction loss $L_{sim}$ is computed between the averaged gradient updates $\nabla\bar{\theta}^c$ computed from the update the client sent and $\nabla \widetilde{\theta}^c$ and which is the same quantity computed from the simulated update in SimUpdate (See Algorithm 2). As we point out in Section 4.3, any reconstruction loss $L_{sim}$ can be used but we rely on the cosine distance from Geiping et al. (2020), resulting in the reconstruction error:
>  $$L_{sim} = 1 - \frac{\langle\nabla\bar{\theta}^c , \nabla \widetilde{\theta}^c\rangle}{||\bar{\theta}^c||_2 || \nabla \widetilde{\theta}^c||_2}$$.
>
> **Q:** In section 4.5 and section 5, what is $U^{c}$, and why would it affect the performance in Table 1 and Table 3?
>
> **A:** $U^c$ is defined in Section 4.3 as $U^c = \mathcal{E} \cdot \mathcal{B}^c$. It represents the number of local weight updates of the client before it shares its update with the server. The higher $U^c$ is, the more unseen local weight updates are executed by the client, which in turn results in harder reconstruction as outlined in Section 4.1.
>
> **Q:** In section 5, how are “the percentage of images that were successfully reconstructed by the attack” computed?
>
> **A:** As described in Section 5.1, we consider an image to be successfully reconstructed on FEMNIST if the PSNR value between the original and reconstructed image (after matching) is > 20 and similarly for CIFAR100 if the PSNR value is > 19. We then compute the percentage of the images which are successfully reconstructed.

---

> > ### Comment · Reviewer_EebW · 2022-07-29
> > **Thanks for the clarification**
> >
> > Thanks for the clarification.
> >
> > I would hope to see more discussion on the complexity of the algorithm. Double backprop can be time consuming, and it is not clear to me how many steps are needed for the convergence in line 6 of algorithm 2. More concretely, how much more computation are needed for attacking FedAvg compared to FedSGD?
> >
> > I am not sure if PSNR is the best metric here, but I also do not have constructive feedback for the authors. Provide some more discussion will be nice while optional.
> >
> > I am generally satisfied with the response and only remaining requirement is the discussion of computational cost.

---

> > > ### Author Response · Authors · 2022-08-08
> > > **Response to "Response to Reviewer EebW Part 1/2"**
> > >
> > > **Q:** Can you provide discussion of the complexity and convergence speed of the double-backprop element of your algorithm compared to FedSGD?
> > >
> > > **A:** Precise theoretical analysis of the complexity and convergence speed of the double-backprop is very hard due to the number of different factors involved in it. This includes factors like learning rate, learning rate decay, computational complexity of calculating the second order derivative etc. To this end, it is beyond the scope of this paper. Instead, below we provide a comparison of the total runtimes obtained in practice for all 100 reconstructions on the FEMNIST experiments originally presented in Table 1 in our paper. All experiments use the same number of iterations ( namely 200 ). As expected, the methods based on simulation are several times slower to execute. The FedSGD-Epoch method presents an interesting tradeoff between reconstruction results and speed as its runtime is much closer to that of the pure FedSGD method while still being able to reconstruct a big portion of the images in many settings. We want to emphasize that in practice the attacker will store the client’s FedAvg updates locally and only later mount the attack outside of the FedAvg learning process. Thus, the time taken to reconstruct the client data is less important than obtaining the data accurately, provided that the attack itself can be executed in a reasonable time.
> > >
> > > $$
> > >   \begin{array}{ccccrrrrr}
> > >      \mathcal{E} & m & N^c & U^c & \text{Ours(prior)} & \text{Ours(no prior)} & \text{Shared} & \text{FedSGD-Epoch} & \text{FedSGD} \\\\
> > >     \hline
> > > 1 & 5  & 50 & 10 & 0.5 \text{ h} & 0.5 \text{ h} & 0.5\text{ h} & 0.25 \text{ h} & 0.25 \text{ h} \\\\
> > > 5 & 5  & 50  & 50 & 2.25 \text{ h} & 2.25 \text{ h} & 2\text{ h} & 0.75 \text{ h} & 0.25 \text{ h}  \\\\
> > >     10 & 10  & 50 & 50 & 2.5 \text{ h} & 2.5 \text{ h} & 2.5\text{ h} & 1.25 \text{ h} & 0.25 \text{ h}  \\\\
> > >   10 & 5  & 50  & 100 & 4.25 \text{ h} & 4.25 \text{ h} & 4.5\text{ h} & 1.5 \text{ h} & 0.25 \text{ h}  \\\\
> > >    10 & 1  & 50  & 500 & 7.5 \text{ h} & 7.5 \text{ h} & 7.5\text{ h} & 1.25 \text{ h} & 0.5 \text{ h}  \\\\
> > >   \end{array}
> > > $$
> > >
> > >
> > > **Q:** Why are you using PSNR, as a metric?
> > >
> > > **A:** We agree with the reviewer that other metrics might be better suited for measuring the reconstruction quality of images resulting from data/gradient leakage in particular. Despite this, PSNR has few advantages that justify our choice of employing it. In particular, it is popular for measuring the quality of image reconstructions in general while also being simple to compute and most previous works in the gradient leakage literature have already employed it in their experiments which allows us to ensure fair comparison. If the reviewer wants a particular metric included in our results, we can supply this in our supplementary in the next revision of this work.

---

> ### Author Response · Authors · 2022-07-13
> **Response to Reviewer EebW Part 2/2**
>
> **Q:** Does your method assume the model updates from each individual client can be observed?
>
> **A:** Yes, throughout the paper we assumed that the server observes the individual gradient updates from each participating client separately. This is a standard assumption in the gradient leakage literature (Geiping et al., 2020; Geng et al., 2021; Yin et al., 2021). Further, designing attacks that work well on aggregated updates is orthogonal to our work and we believe future work that improves the reconstruction from aggregated updates should be easy to incorporate in our method. Finally, as the aggregation happens at the clients, some of the clients will observe individual client updates which in turn will allow these clients to use our attack to reconstruct the data from the updates. In a setting where clients compete with each other or strong privacy guarantees must be enforced (e.g due to legal requirements) such data leakage between clients remains problematic. We nevertheless present results from using our attack on aggregated updates in the next question.
>
>
> **Q:** What if only aggregated results from multiple clients can be observed?
>
> **A:** We experimented with reconstructing images from aggregated updates below. The experiment was carried out on the FEMNIST dataset with individual client updates consisting of 10 epochs with 10 batches of 5 images each. The aggregated updates were computed as the mean of the participating clients’ updates whose count is shown in the first column of the table below. Thus, the first row in the table below, that shows the results of using only 1 client, represents the attack presented in Table 1 in our paper. As in our paper, we report the average reconstruction results across 100 different aggregated client updates. As in Table 1 from our paper, we assume the total label counts for each client are known, but the counts per batch are unknown. To account for the aggregated updates, we changed SimUpdate to simulate all client updates separately with the procedure described in Section 4.3 and then average them to produce the final simulated aggregated update used in Algorithm 2. From the results, we see that as more client updates are aggregated we obtain lower reconstruction rates for all of the attacks. However, we see that our attack method reconstructs the most images in all settings compared to the alternatives, while achieving higher PSNR. Further, our method is capable of reconstructing around 30\% of the images when updates are aggregated across 4 different clients, suggesting that aggregated updates are vulnerable when aggregation with a small number of clients is used. We will include this experiment in the Appendix in the next revision.
> $$
>   \begin{array}{rllllllllll}
>     & \hspace{1cm} \rlap{\text{Ours(prior)}} && \hspace{0.6cm} \rlap{\text{Ours(no prior)}}  && \hspace{1.4cm} \rlap{\text{Shared}} &&  \hspace{0.4cm} \rlap{\text{FedSGD-Epoch}}  && \hspace{1cm} \rlap{\text{FedSGD} }\\\\
> & \rlap{\underline{\hspace{3.6cm}}} &&  \rlap{\underline{\hspace{3.6cm}}}  &&  \rlap{\underline{\hspace{3.6cm}}} &&  \rlap{\underline{\hspace{3.6cm}}} && \rlap{\underline{\hspace{3.6cm}}}\\\\
>     \text{\\# clients} & \quad Rec(\\%) \quad & PSNR & \quad Rec(\\%) \quad & PSNR & \quad Rec(\\%) \quad & PSNR & \quad Rec(\\%)
>  \quad & PSNR & \quad Rec(\\%) \quad & PSNR \\\\
>     \hline
>     1 & \hspace{0.8cm} \textbf{65.5} & \hspace{0.3cm}  \textbf{21.5} &  \hspace{0.8cm} 50.6 & \hspace{0.3cm} 20.3 &  \hspace{0.8cm} 59.9 & \hspace{0.3cm} 20.3 &  \hspace{0.8cm} 56.0 & \hspace{0.3cm} 19.6 &  \hspace{0.8cm} 8.3 & \hspace{0.3cm} 16.7 \\\\
>     2 & \hspace{0.8cm} \textbf{42.1} & \hspace{0.3cm}  \textbf{19.8} &  \hspace{0.8cm} 32.3 & \hspace{0.3cm} 18.7 &  \hspace{0.8cm} 27.6 & \hspace{0.3cm} 18.3 &  \hspace{0.8cm} 25.7 & \hspace{0.3cm} 17.7 &  \hspace{0.8cm} 6.8 & \hspace{0.3cm} 16.1  \\\\
>     4 &  \hspace{0.8cm}  \textbf{29.7} & \hspace{0.3cm}  \textbf{18.8} &  \hspace{0.8cm} 24.8 & \hspace{0.3cm} 17.8 &  \hspace{0.8cm} 14.8 & \hspace{0.3cm} 17.0 &  \hspace{0.8cm} 14.3 & \hspace{0.3cm} 16.3 &  \hspace{0.8cm} 5.7 & \hspace{0.3cm} 15.5
>   \end{array}
> $$
>
> **Q:** I would encourage the authors to discuss limitations and possible defenses of the proposed method.
>
> **A:** We have provided experiments and discussion on the effectiveness of our method on trained and defended networks in Appendix B.1 and B.2 respectively, where we demonstrated that adding Gaussian noise to gradients can result in much less reconstructed images (even though around 15% of the images are still reconstructed) in exchange for network accuracy. Further, as we point out in Section 5, as $U^c$ grows larger, our attacks become less effective and this can be used as an effective defense in exchange for increased client computation. We further provided broader impact discussion in Appendix D. We will better emphasize these shortcomings of our work in the next revision of the paper.

---

> > ### Comment · Reviewer_EebW · 2022-07-29
> > **Thanks for the experiments, acknowledge the current status of the literature, disagree with "the aggregation happens at the clients"**
> >
> > Thanks for the experiments.
> >
> > I acknowledge previous work often assume individual client updates can be observed, and I think it is fine to follow the literature and leave aggregated results as future work. That being said, I would encourage authors to properly discuss this. There are papers start looking at the aggregated results, for example, [Fishing for User Data in Large-Batch Federated Learning via Gradient Magnification].
> >
> > Also, for FedSGD, the effective batch size is probably the most important factor for aggregated results; for FedAvg, since each client performs local update, this changes the "aggregation".
> >
> > I do not understand and disagree with the claim "the aggregation happens at the clients".

---

> > > ### Author Response · Authors · 2022-08-08
> > > **Response to "Response to Reviewer EebW Part 2/2"**
> > >
> > > Thank you for pointing out our mistake in the answer above. Following our discussion, we have provided the aggregation experiment above in Section B.5 in the latest revision of the paper.
> > >
> > > **Q:** I would encourage authors to properly discuss that they are not considering aggregation results.
> > >
> > > **A:** We have emphasized that reconstruction from aggregated updates is not a focus of this work in Section  2 and provided the experiment above in Section B.5 of the supplementary material of the latest version of our paper.
> > >
> > > **Q:** I do not understand and disagree with the claim "the aggregation happens at the clients".
> > >
> > > **A:** We apologize for having made a mistake stating that “the aggregation happens on the clients”. This statement and the point we made with it is indeed invalid. We instead wanted to emphasize that classic secure multi-party computation methods require communication between the parties (here clients), which can be impractical when the number of clients in FL is very large and communication between them is limited.

---

### Review · Reviewer_NCe3 · 2022-07-21

**Summary Of Contributions:**

This paper considers reconstructing private training data of clients by inspecting their submitted model updates (i.e., gradients). Unlike existing works that assume local clients perform FedSGD, this paper considers the more practical and challenging setting of FedAvg, where the local clients performs multiple steps of model updates (number of batches times number of epochs). This paper proposes improved version of label reconstruction and input reconstruction algorithms, coupled with epoch order invariant prior and epoch matching strategy. Empirically, the proposed method achieves state-of-the-art attack performance, compared to existing baselines designed for FedSGD, in various FedAvg settings.

**Broader Impact Concerns:**

The authors addressed the ethical concerns well.

**Requested Changes:**

I still do not understand why the attacker can have exact knowledge on the exact number of local epochs and batches. I think, for FedAvg, what the server needs is their product, which will then be used to do weighted averaging over the submitted model updates. Therefore, I think the paper should perform more experiments when the attacker (server) only knows $U^c$, but not the exact local updates and batches. We mainly want to see how sensitive the proposed solution is to the choice of different local updates and batches during simulation. Although I like the overall idea presented in the paper, I believe this adjustment is critical for securing my recommendation for acceptance.

Presentation wise, it will be helpful to highlight the main difference between the proposed one and existing baselines  (e.g., Geiping et al., (2020), Geng et al., (2021) in terms of input and label count reconstruction by comparing them in a side by side fashion. This way, it can better highlight the major contributions in this paper: linear interpolation of local statistics and grouping optimization variable helps a lot. This will simply strengthen the work and is not critical for acceptance.

**Strengths And Weaknesses:**

Strengths:
1. The paper is written is adequately well and is easy to follow.
2. The design of experiments validates the importance of each building block of the end-to-end attack.
3. The attack performs significantly better than the existing baselines in various FedAvg attack settings.

Weakness:
1. Attacker's knowledge on the exact number of local epochs and batches might be unrealistic.

---

> ### Author Response · Authors · 2022-08-08
> **Response to Reviewer NCe3 Part 1/2**
>
> We thank the reviewer for the insightful comments and questions on the experimental design and the paper’s writing. We address the reviewer’s concerns below.
>
> **Q:**  Is the assumption that the attacker has knowledge on the exact number of local epochs and batches realistic? Is $U^c$ not enough?
>
> **A:** We thank the reviewer for raising this point. We can see a practical setting where only sharing $U^c$ and $N^c$ can be enough for the FedAvg learning process to converge. We note that sharing $N^c$ is needed in order to allow the server to balance the accuracy of the learned network between the clients according to the amount of data they hold. Since $U^c = \mathcal{E} \cdot \mathcal{B}^c$, the attacker knows that $\mathcal{E}$ is a divisor of $U^c$. Further, for a given choice of $\mathcal{E}$, the attacker can calculate $\mathcal{B}^c$ by the formula $\mathcal{B}^c = \frac{U^c}{\mathcal{E}}$. Assuming that all batches used in the client are of the same size (except maybe the last one), the attacker can obtain the value of $m$ by the formula $m = \lceil \frac{N^c}{\mathcal{B}^c} \rceil$ for each $\mathcal{E}$. If the attacker has access to the value of $U^c$, separate reconstructions with all divisors of $U^c$ substituted for $\mathcal{E}$ can be executed by first computing the corresponding value of $m$ and then for each pair executing our attack. The attacker can simply choose the best of these reconstructions by visually inspecting the results. Additionally, knowing some side-channel information about the client device on which the updates are computed, e.g maximum allowed RAM or GPU usage can further shorten the list of possible candidates for $m$ and thus for $\mathcal{E}$. In the case where visual inspection is not possible ( e.g due to the nature of the data being reconstructed ), we demonstrate that our method is still capable of reconstructing a big portion of the data in Part 2 of our answer.
>
> **Q:** It will be helpful to the presentation of the paper if you outline the differences between the proposed method and existing baselines in the paper.
>
> **A:** Thank you for the suggestion, we will expand this discussion in the next revision of the paper.

---

> ### Author Response · Authors · 2022-08-08
> **Response to Reviewer NCe3 Part 2/2**
>
> **Q:** Can you provide an experiment to demonstrate how sensitive the proposed solution is to the choice of different local updates and batches during simulation when $U^c$ is known?
>
> **A:** We experimented with reconstructing data when only $U^c$ and $N^c$ are available to the attacker but not $\mathcal{E}$ and $m$. To this end, we generate FedAvg client updates with $\mathcal{E}=5$, $m=5$ and $N^c=50$ on the FEMNIST dataset and then reconstruct data from these updates by plugging different values of $\mathcal{E}$ and $m$ in our algorithm. As described in Part 1 of our answer, it is known that $\mathcal{E}$ is a divisor of $U^c=50$. To this end, we choose $\mathcal{E}$ to be either $10$, $25$ or $50$, resulting in values for $m$ of $10$, $25$ and $50$, respectively. The results of reconstructing the data with these (wrong) choices of the parameters $\mathcal{E}$ and $m$ for the methods in Table 1 in the paper and known labels are presented in the table below. In the table below, we also present the reconstruction with the correct values of $\mathcal{E}$ and $m$ for comparison. The ‘True’ columns represent the value used by the clients to compute their FedAvg update, while the ‘Chosen’ columns represent the values used by the algorithms during reconstruction. We see several patterns in the results. First, the image reconstruction becomes worse the further the chosen value of $\mathcal{E}$ is from the correct value $5$ for all methods. Despite this, our method still outperforms the baselines and it is capable of reconstructing more than $50\\%$ of the images in all cases. Second, our order-invariant prior is less effective when the true value of $\mathcal{E}$ is unknown. We think there are two reasons for that. One is that the same value of $\lambda_{\text{inv}}$ chosen on the correct values of $\mathcal{E}$ and $m$ is used in all experiments which can be suboptimal when $\mathcal{E}$ is chosen wrongly. The other is that our prior relies on the knowledge of the correct number of epochs $\mathcal{E}$ and therefore is less effective when the wrong value of $\mathcal{E}$ is used. Finally, unlike our method, we see that the performance of the shared variable method degrades drastically when the choice of $\mathcal{E}$ is wrong. This is in line with our other experiments where we also see this method is less robust with respect to changes to the FedAvg client updates. We have added these results and discussion to Section B.4 in the Appendix of our paper.
> $$
>   \begin{array}{cccccccccccccccccc}
>     \hspace{2cm} \rlap{\mathcal{E}} && \hspace{2cm} \rlap{m} && \hspace{2cm} \rlap{N^c} && \hspace{2cm} \rlap{U^c} && \hspace{1cm} \rlap{\text{Ours(prior)}} && \hspace{0.6cm} \rlap{\text{Ours(no prior)}}  && \hspace{1.4cm} \rlap{\text{Shared}} &&  \hspace{0.4cm} \rlap{\text{FedSGD-Epoch}}  && \hspace{1cm} \rlap{\text{FedSGD} }\\\\
> \hspace{-0.8cm}\rlap{\underline{\hspace{3.2cm}}} && \hspace{-0.8cm}\rlap{\underline{\hspace{3.2cm}}} && \hspace{-0.8cm}\rlap{\underline{\hspace{3.2cm}}} && \hspace{-0.8cm}\rlap{\underline{\hspace{3.2cm}}} && \hspace{-0.8cm}\rlap{\underline{\hspace{3.6cm}}} &&  \hspace{-0.8cm}\rlap{\underline{\hspace{3.6cm}}}  &&  \hspace{-0.8cm}\rlap{\underline{\hspace{3.6cm}}} &&  \hspace{-0.8cm}\rlap{\underline{\hspace{3.6cm}}} && \hspace{-0.8cm}\rlap{\underline{\hspace{3.6cm}}}\\\\
>     \\  True & Chosen & \\  True & Chosen & \\  True & Chosen & \\  True & Chosen & \hspace{1.2cm} Rec(\\%) \quad & PSNR & \hspace{1.2cm} Rec(\\%) \quad & PSNR & \hspace{1.2cm} Rec(\\%) \quad & PSNR & \hspace{1.2cm} Rec(\\%)
>  \quad & PSNR & \hspace{1.2cm} Rec(\\%) \quad & PSNR \\\\
>     \hline
>     5 & 5 & 5 & 5 & 50 & 50 & 50 & 50 & \hspace{0.8cm} \textbf{63.1} & \hspace{0.3cm}  \textbf{21.2} &  \hspace{0.8cm} 56.2 & \hspace{0.3cm} 20.7 &  \hspace{0.8cm} 59.0 & \hspace{0.3cm} 20.6 &  \hspace{0.8cm} 56.1 & \hspace{0.3cm} 20.4 &  \hspace{0.8cm} 13.7 & \hspace{0.3cm} 17.4 \\\\
>     5 & 10 & 5 & 10 & 50 & 50 & 50 & 50 & \hspace{0.8cm} \textbf{55.8} & \hspace{0.3cm}  \textbf{20.7} &  \hspace{0.8cm} 55.2 & \hspace{0.3cm} 20.6 &  \hspace{0.8cm} 53.8 & \hspace{0.3cm} 20.1 &  \hspace{0.8cm} 53.8 & \hspace{0.3cm} 20.1 &  \hspace{0.8cm} 12.7 & \hspace{0.3cm} 17.3 \\\\
>    5 & 25 & 5 & 25 & 50 & 50 & 50 & 50 & \hspace{0.8cm} 51.0 & \hspace{0.3cm}  20.2 &  \hspace{0.8cm} \textbf{51.7} & \hspace{0.3cm} \textbf{20.3} &  \hspace{0.8cm} 40.5 & \hspace{0.3cm} 19.0 &  \hspace{0.8cm} 39.7 & \hspace{0.3cm} 19.0 &  \hspace{0.8cm} 11.4 & \hspace{0.3cm} 17.1  \\\\
>     5 & 50 & 5 & 50 & 50 & 50 & 50 & 50 & \hspace{0.8cm}  \textbf{52.0} & \hspace{0.3cm}  \textbf{20.2} &  \hspace{0.8cm} 51.4 & \hspace{0.3cm} 20.2 &  \hspace{0.8cm} 34.3 & \hspace{0.3cm} 18.5 &  \hspace{0.8cm} 34.6 & \hspace{0.3cm} 18.5 &  \hspace{0.8cm} 10.8 & \hspace{0.3cm} 17.0
>   \end{array}
> $$

---

### Review · Reviewer_JRGU · 2022-07-28

**Summary Of Contributions:**

The authors describe how existing attacks against FL pipelines do not operate in the more realistic setting of FedAvg wherein multiple local update steps are taken before parameter updates are shared with the central server. Many existing attacks instead assume a simplified setting where a single local update is calculated and shared. The authors then describe several adaptations/additions to data leakage attacks that improve their operation in the FedAvg setting.

**Broader Impact Concerns:**

I do not have any concerns about the ethical implications of the work.

**Requested Changes:**

I propose a few main adjustments

- Please add more detail, explanation, and motivation for the epoch order-invariant prior. I would expand section 4.4, as well as 5.4. (critical)
- Please explain what "Rec %" means in your tables. How are you measuring the number of recovered images? (recommended)
- Include experiments on ImageNet (recommended)

**Strengths And Weaknesses:**

**Strengths**
- The paper is well motivated - existing attacks usually do operate in limited settings.
- Presentation and writing is generally clean and easy to follow.
- The paper proposes several novel approaches that are well motivated and successful (compared with the baseline)

**Weaknesses**
- The exposition of the epoch order-invariant prior could use some improvement. It's not completely clear to me what purpose this serves/the motivation for this additional loss term.
- Experimentation is on "simple" datasets. I realize that resources could be limited, but it really would have been nice to include some results on a more complex dataset such as ImageNet. Previous data leakage attacks have done this, and I'm quite curious about the reconstruction quality in the FedAvg scenario for higher resolution inputs.

---

> ### Author Response · Authors · 2022-08-08
> **Response to Reviewer JRGU Part 1/2**
>
> We thank the reviewer for the questions and for acknowledging the novelty of our approaches. We address the reviewer’s concerns below.
>
> **Q:** Can you further discuss the motivation behind your epoch order-invariant prior?
>
> **A:**  As demonstrated in [1], exploiting prior information about the reconstructed inputs in the form of additional error terms in the reconstruction loss can result in better overall data leakage results. To this end, we introduce our epoch order-invariant prior that exploits the prior information that the same set of images are being used at each epoch to compute the FedAvg client update. Recall from Section 4.3 that our method uses separate reconstruction variables for reconstructing the different copies of the same image at different client epochs. Ideally,  we want to enforce the property that all of the reconstruction variables corresponding to the same image hold similar values using our reconstruction loss. However, the order of the reconstructed images is not consistent across epochs as the client randomly splits the images into different sets of batches at each epoch when computing its update. Further, how the order changes across the epochs is not known by the server. To this end, we cannot directly enforce the desired property as we don’t know which images correspond to each other at the different epochs during the reconstruction process. Instead, we can only enforce a weaker property that a function $g$ that takes as inputs all of the images within an epoch has the similar value at different epochs provided that $g$ is order-invariant (e.g $g(a,b) = g(b,a)$, where $a$ and $b$ are the only two images at a given epoch). An example of such an order-invariant function $g$ is calculating the pixelwise mean of all of the images in an epoch. Equation 3 from the paper naturally captures this weaker property by computing the distance between the values of $g$ computed at every pair of epochs. In Section 5.4, we investigate different order-invariant functions $g$ inspired by previous work on order-invariant functions in neural networks presented in [2], as well as different notions of distance $D\_{\text{inv}}$ between the values of $g$. We hope this discussion clarifies the existing motivation and explanation of our prior already provided in our paper. Following your suggestion, we plan to expand Section 4.5 and Section 5.4 in the next revision of our paper. We kindly ask the reviewers to list particular points that were unclear in the original description to allow us to concentrate our efforts on these parts of the writing.
>
> **Q:** What do the "Rec %" columns represent in your tables? How are you measuring the number of recovered images in your experiments?
>
> **A:** As described in Section 5.1, we consider an image to be successfully reconstructed on FEMNIST if the PSNR value between the original and reconstructed image (after matching) is > 20 and similarly for CIFAR100 if the PSNR value is > 19. The "Rec %" columns represent the percentage of all client images that were successfully reconstructed by the different algorithms.

---

> ### Author Response · Authors · 2022-08-08
> **Response to Reviewer JRGU Part 2/2**
>
> **Q:** Can you include results on more complex datasets such as ImageNet, similarly to previous gradient leakage attacks?
>
> **A:** Any interesting FedAvg experiment implicitly requires that the simpler setting of reconstructing images from FedSGD updates computed on a single batch of size $N^c$ works, as the FedAvg updates computed on the same client dataset will be even harder to attack. Further, any interesting FedAvg experiment requires that not too small values of $N^c$ are used to allow for experiments with different sizes of $m$. While many FedSGD attacks have demonstrated some results on ImageNet, they often focus on the simple setting where the batches contain a very small number of images (e.g [4] experiments with batch size of 1). As demonstrated by [3], this is due to bad results on the bigger batch sizes unless very advanced priors are used. The two priors we are aware of that can fix these issues and allow good reconstructions on large batch sizes of ImageNet images are the batch norm prior demonstrated by [3] and the GAN prior demonstrated by [5]. The implementation of both of these priors is highly non-trivial. Further, searching for good hyperparameters for these attacks is outside of our computational capabilities. For these reasons we consider ImageNet experiments outside of the scope of our work.
>
> [1] Mislav Balunović, Dimitar I Dimitrov, Robin Staab, and Martin Vechev. Bayesian framework for gradient leakage. arXiv preprint arXiv:2111.04706, 2021.
>
> [2] Charles R Qi, Hao Su, Kaichun Mo, and Leonidas J Guibas. Pointnet: Deep learning on point sets for 3d classification and segmentation. In Proceedings of the IEEE conference on computer vision and pattern recognition, pp. 652–660, 2017.
>
> [3] Hongxu Yin, Arun Mallya, Arash Vahdat, Jose M Alvarez, Jan Kautz, and Pavlo Molchanov. See through gradients: Image batch recovery via gradinversion. In Proceedings of the IEEE/CVF Conference on Computer Vision and Pattern Recognition, pp. 16337–16346, 2021.
>
> [4] Jonas Geiping, Hartmut Bauermeister, Hannah Dröge, and Michael Moeller. Inverting gradients-how easy is it to break privacy in federated learning? Advances in Neural Information Processing Systems, 33: 16937–16947, 2020.
>
> [5] Zhuohang Li, Jiaxin Zhang, Luyang Liu, and Jian Liu. Auditing privacy defenses in federated learning via generative gradient leakage. arXiv preprint arXiv:2203.15696, 2022.

---

### Decision · Action_Editors · 2022-09-15

**Recommendation:** Accept with minor revision

**Comment:**

Reviewers raised important points of concern and clarification on the details, but were overall happy with the direction of the paper. It has been a couple of years since the first work on data leakage from FedSGD. The time is ripe for exploring leakage in more practical scenarios with FedAvg. As this is one of the first papers to specifically address this practical discrepancy, the motivation of the paper is highly relevant. The authors come up with a novel epoch order invariant prior and epoch matching strategy and to perform input and label reconstruction and perform a robust evaluation with FedSGD as the baseline. Reviewers did not take issue with the correctness of the evaluations and mostly focused their concerns on presentation, clarification, or further experiments.

Experiments were performed on small datasets, FEMNIST and CIFAR100, making it harder to judge reconstruction quality. One reviewer requested larger scale (ImageNet) experiments. While this would have made the paper's claims more clear, I don't think it is a show-stopper. ImageNet experiments with 2nd order gradients would indeed require significantly more resources (authors mentioned they're using RTX2080 cards which are for typically for those on a smaller budget), and I think it's reasonable to allow having only small dataset experiments as long as the evaluation is good enough and the ideas are novel.

Another reviewer raised an importance concern that the attacker's knowledge of the epoch count and batch size is too impractical, and suggested that knowledge of only the total number of steps (U^c) is a more realistic situation. The authors were able to respond with experiments in this more realistic scenario and show that the attack remains pretty robust (certainly better than FedSGD) when the attackers guess wrongly the epoch count and batch size. Further, the reviewer that brought this up, NCe3, also agreed that they were satisfied with the authors' response.

Finally, yet another reviewer had the concern about computational cost of this attack. The authors provided new data showing that the computational cost measured via runtime is about 10x higher than the baseline. This info is certainly good to know for downstream analysis of this threat vector, and I see it as something that makes the paper more thorough. I don't see the fact that it takes more compute as disqualifying and it seems the reviewer that brought it up also doesn't think so given their strong recommendation of the paper.

Lastly, reviewers were in agreement that the ethical concerns were handled properly in this paper.

Overall I think this is a good paper and recommend Accept with minor revision.

Minor revisions required:
- remove claims that “the aggregation happens on the clients”
- improve clarity of Algorithm 2 and surrounding text
- incorporate all clarification statements and new data from this rebuttal process into the manuscript (appendix section if space-limited for the data)

---

> ### Author Response · Authors · 2022-11-01
> **Re: Decision**
>
> We thank the reviewers and Action Editor for their comments. We have addressed them in our Camera-ready revision. Let us know if any additional changes need to be incorporated.